# Microclot array elastometry for integrated measurement of thrombus formation and clot biomechanics under fluid shear

Zhaowei Chen [1], Jiankai Lu[1], Changjie Zhang[2], Isaac Hsia[1], Xinheng Yu [2], Leo Marecki[1], Eric Marecki[1], Mohammadnabi Asmani [1], Shilpa Jain[3], Sriram Neelamegham[2] & Ruogang Zhao [1]

Blood clotting at the vascular injury site is a complex process that involves platelet adhesion and clot stiffening/contraction in the milieu of fluid flow. An integrated understanding of the hemodynamics and tissue mechanics regulating this process is currently lacking due to the absence of an experimental system that can simultaneously model clot formation and measure clot mechanics under shear flow. Here we develop a microfluidic-integrated microclot-array-elastometry system (clotMAT) that recapitulates dynamic changes in clot mechanics under physiological shear. Treatments with procoagulants and platelet antagonists and studies with diseased patient plasma demonstrate the ability of the system to assay clot biomechanics associated with common antiplatelet treatments and bleeding disorders. The changes of clot mechanics under biochemical treatments and shear flow demonstrate independent yet equally strong effects of these two stimulants on clot stiffening. This microtissue force sensing system may have future research and diagnostic potential for various bleeding disorders.

[1] Department of Biomedical Engineering, State University of New York at Buffalo, Buffalo, NY 14260, USA. [2] Department of Chemical and Biological Engineering and Clinical & Translational Research Center, State University of New York at Buffalo, Buffalo, NY 14260, USA. [3] Hemophilia Center of Western New York, Department of Pediatrics, Division of Pediatric Hematology-Oncology, John R. Oishei Children's Hospital of Buffalo, Buffalo, NY 14203, USA. Correspondence and requests for materials should be addressed to S.N. (email: neel@buffalo.edu) or to R.Z. (email: rgzhao@buffalo.edu)

**B**lood clotting at the vascular injury site is a complex biomechanical process mediated by several key factors including platelet deposition and contraction, clot stiffening and hemodynamic forces[1,2]. In hemostasis, the interplay of these mechanical factors leads to strong clots that stem bleeding, but in clotting disorders such as von Willebrand disease (VWD), disrupted biomechanical interactions lead to weak clots and bleeding[3–5]. Although different aspects of the clotting problem such as shear flow-mediated platelet adhesion and single platelet mechanics have been studied in the past[6,7], an integrated understanding of the interplay between clot mechanics and hemodynamic shear is still missing. This is mainly due to the lack of combined capacity to model and measure clotting mechanics in the presence of shear flow in existing research models.

Recent technological advancement has resulted in the evolution of independent approaches that focus on different parts of the clotting problem. For example, microfluidics systems create well-controlled blood flow in matrix protein-coated microchannels, and thus they have become a widely used vascular injury model for the study of platelet adhesion, aggregation and thrombus formation[8]. These systems are powerful tools to study the effects of various flow conditions, surface chemistries and coagulation factors on platelet adhesion and aggregation[9–13]; however, they are not equipped with the ability to measure clot mechanics. Furthermore, matrix proteins are coated on glass or relatively rigid surfaces, which do not mimic the physiological stiffness of exposed matrix at vascular injury sites. In the study of platelet mechanics, atomic force microscopy and flexible micropost arrays have been used to measure single platelet-generated contractile forces under static conditions[2,14], and matrix protein microdot arrays have been used to study the effects of substrate stiffness and mechanosensitive signaling pathways on platelet-generated contractile force[15,16]. Single platelet-mediated fibrin fiber remodeling has also been studied using confocal microscopy[17]. While these studies provide important information on platelet mechanics, they do not include the effect of hemodynamic forces and offer limited insight into clot mechanics at the tissue level. In the clinic, cup and cone type of rheometers, also known as thromboelastography (TEG) or rotation thromboelastometry (ROTEM), have been developed to measure clot stiffness at different stages of the coagulation process[18,19]. Although these systems provide useful information that can assist the diagnosis of coagulation disorders, their consideration of hemodynamic flow and platelet−matrix interactions is inadequate. As a result, their diagnostic power is limited and they are seldom applied for disorders of primary hemostasis, like VWD[18,19]. Together, the lack of an integrated system that can follow clot formation and mechanical properties in the presence of shear flow has hindered an integrated understanding of the biomechanical events occurred during blood clotting.

With recent advent of microfabrication technology, we and others have created microtissue array-based mechanical sensing platforms that allow simultaneous control of tissue formation and measurement of tissue mechanics[20–22]. In this system, cell contraction-mediated self-assembly of matrix proteins such as collagen and fibrin leads to the formation of submillimeter microtissues that anchor on top of multiple flexible poly(dimethylsiloxane) (PDMS) micropillars. These micropillars serve as mechanical sensors that can report in-situ tissue mechanical properties. The composition and stiffness of these microtissues, contributed both by preseeded and cell-secreted matrix proteins, mimic those of native vascular tissue. By virtue of their small dimensions, the microtissues offer orders of magnitude scale-up advantages over conventional bulk hydrogel models and open up the possibility to integrate with microfluidics. This system has been used to study the mechanics of cardiac and skeletal muscles,

fibrotic tissues and wound healing[21,23–27]; however, it has not been integrated with the shear flow.

In the current study, we develop a microfluidic-integrated microclot array elastometry (clotMAT) system by integrating a collagen microtissue array-based mechanical sensing platform with platelet flow. This system recapitulates the dynamic process of platelet adhesion and clot formation under various flow conditions and reports changes in clot mechanical properties in real time. Through platelet agonist (ADP and thrombin) treatments, we dissected the contributions of hemodynamic and tissue/cell forces to clot stiffening. Treatments with platelet antagonist (glycoprotein receptor antibodies, blebbistatin and acetylsalicyclic acid (ASA)) further enabled the examination of the effects of platelet adhesion and cytoskeletal tension on clot mechanics. The utility of the clotMAT system in bleeding disorders was demonstrated through the study of clotting mechanics of VWD. Integrated analysis of results show strong correlation between clot stiffness, clot contractile force and the level of clot retraction (volume shrinkage) under both biochemically treated conditions and shear flow conditions, suggesting these two stimulants independently yet equally strongly affect clot remodeling and stiffening. This finding unveils the tissue remodeling/stiffening mechanism in a previously unexplored shear flow regime, and thus it significantly expands the boundary of previously established tissue remodeling theory. Together, this work describes a physiologically relevant microtissue force sensing system for the study of normal and abnormal clot biomechanics under shear flow, which may have future research and diagnostic potential for various bleeding disorders.

## Results

**Microclot array formation under platelet flow.** The clotMAT device consists of three functional layers including a top PDMS layer containing multiple microfluidic channels for platelet flow, a middle PDMS layer containing arrays of collagen microtissues that capture flowing platelets and report clot contractile forces, and a bottom stretchable silicone membrane for clot stiffness measurement (Fig. 1a, b). The middle and bottom layers were bonded through the formation of irreversible covalent bonds upon casting liquid PDMS on silicone membrane, and the top and middle layers were bonded temporarily through sustained pressure (Supplementary Fig. 1). We fabricated the middle PDMS layer using a multilayer photolithography and soft-lithography technique as previously described[21,28]. An array of submillimeter-size microwells (13 columns by 6 rows) were cast in the middle PDMS layer, and each microwell contains a pair of flexible PDMS micropillars that serve as force sensors to report clot-generated contractile forces (Fig. 1a, b). A single micropillar consists of a slim leg section and an enlarged head section that provides mechanical constraint to the suspended microtissue (Supplementary Fig. 2). The collagen microtissue array was fabricated by utilizing endothelial cell-mediated collagen gel compaction (Supplementary Movie 1)[21]. Additional steps were performed to remove originally seeded endothelial cells after dog-bone-shaped microtissue formation (Supplementary Fig. 1), in order to recapitulate the matrix composition anticipated during vascular injury. The resulting individual bare micro-collagen matrix hangs between the heads of a pair of flexible micropillars (Supplementary Figs. 2, 3), mimicking exposed subendothelial collagen at vessel injury sites and avoiding the effect of rigid substrates used in existing microfluidic devices[12]. The bare collagen microtissues measure $132.7 \pm 17.2\ \mu m$ (mean ± SD, 113 replicates) wide and $80.7 \pm 5.9\ \mu m$ (16 replicates) thick. The microfluidic channels in the top PDMS layer were aligned with individual rows of collagen microtissues in the middle layer (Supplementary Fig. 4).

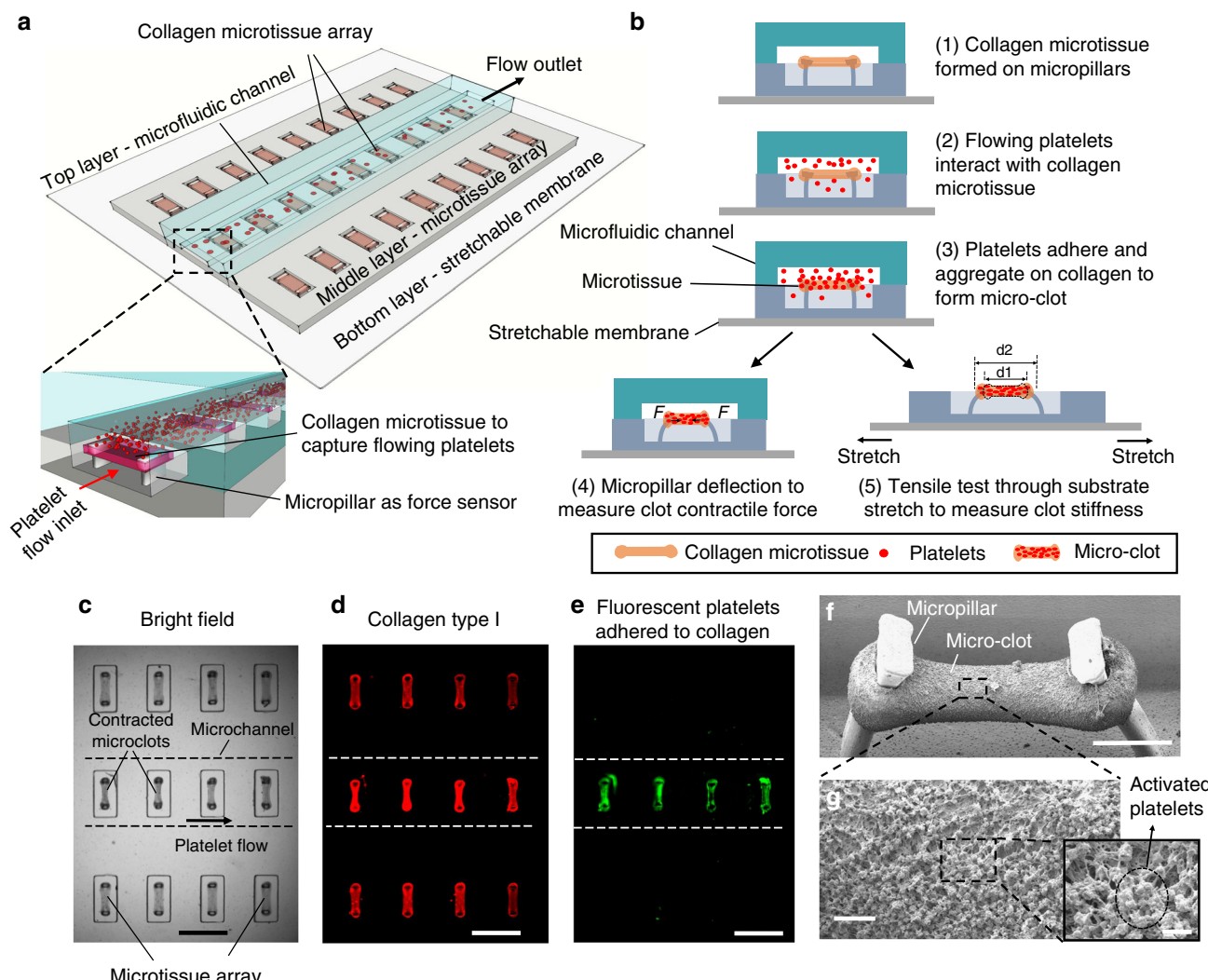

**Fig. 1** clotMAT system setup and microclot array formation under platelet flow. **a** clotMAT system contains a top microfluidic channel layer, a middle PDMS substrate containing the microtissue array and a bottom stretchable membrane. An array of exposed collagen microtissues capture flowing platelets to form individual microclots. **b** Schematic diagram shows microclot formation process and the principles of mechanical property measurement. Microclot contractile force is measured by micropillar deflection and stiffness is measured by substrate stretching-enabled tensile testing. A region of the microtissue array shown by bright field image (**c**), and immunofluorescence staining of the collagen (**d**) and adhered platelets (**e**) in the same region. Scale bar is 800 μm. Note that the platelet flow ran through the middle row of the collagen microtissue array, resulting in the adhesion of fluorescent-labeled platelets (green) only in this row. **f** SEM image of a microclot formed between the heads of a pair of micropillars. Obvious micropillar deflection is noticeable. Scale bar is 200 μm. **g** Zoom-in view of the microclot surface shows numerous platelets trapped in a fibrin meshwork. Scale bar is 10 μm. Inset shows activated single platelets. Scale bar is 2 μm. PDMS poly(dimethylsiloxane), SEM scanning electron microscopy

When the microchannels were perfused with citrated platelet-rich plasma (PRP), flowing platelets were arrested on a row of individual micro-collagen matrices contained by the microchannel (Fig. 1c−e). The adhered platelets gradually covered the collagen matrix, as demonstrated by the gradual increase in fluorescence intensity of the microtissue (Supplementary Movie 2). Over a period of 30 min, platelets continued to aggregate and fibrin continued to polymerize over the collagen matrix surface, leading to the formation of a submillimeter-sized microclot hanging between two micropillars (Fig. 1f, g). SEM and confocal imaging of microclots formed using healthy PRP show that aggregated platelets and fibrin meshwork together formed a thick clot shell covering the bare collagen core. The width of the whole microclot is 83.3 ± 9.2 μm (14 replicates). The thickness of the clot shell is 28.2 ± 7.6 μm (14 replicates) and that of the whole clot is 64.1 ± 5.8 μm (14 replicates, Supplementary Figs. 5–7). Compared to the bare collagen matrix before flow, the size of the collagen core decreased after platelet flow. Such compaction of the matrix is contributed by the collective contraction of activated platelets, as detected by the deflection of the micropillars (Fig. 1f, g).

**Modeling clot contraction and stiffening by microclots.** Upon clot formation, activated platelets generate contractile forces that lead to clot retraction[29]. To detect the minute change in microclot-generated contractile forces, micropillars were engineered to sense force in nanoNewton range (spring constant $k = 120$ nN μm$^{-1}$, Supplementary Fig. 2), and micropillar head displacements ($\delta1$ and $\delta2$) were used to calculate contractile force according to cantilever bending theory (Fig. 2a). Here, we observed that micropillars were sensitive enough to detect the increase in clot contractile force upon platelet flow, as demonstrated by the obvious inward deflection of the micropillars (Figs. 1f, 2b, Supplementary Fig. 8). Such increase in contractile force is

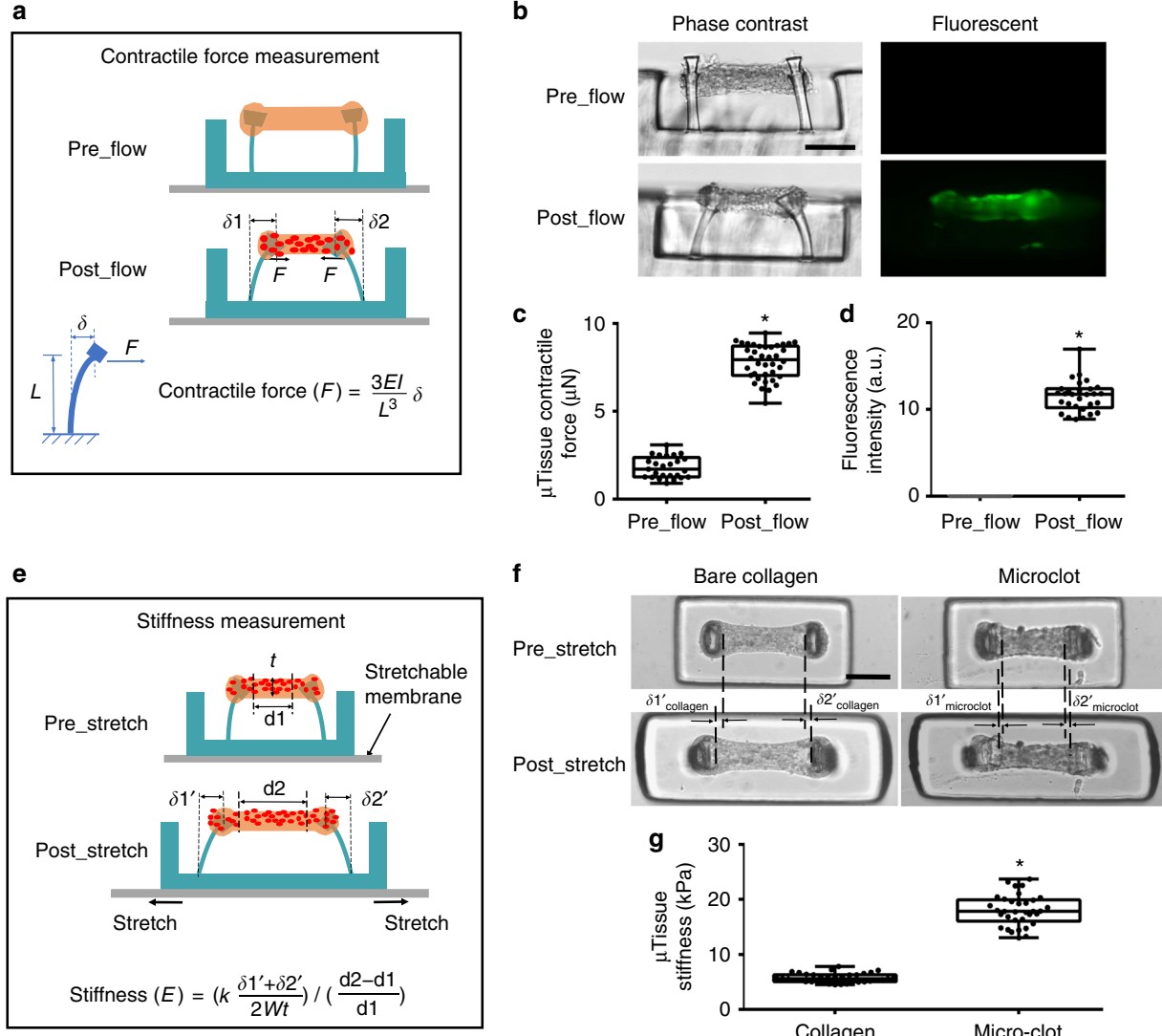

**Fig. 2** Measurement of clot contraction and stiffening in the microclot array. Schematic diagram (**a**) and phase contrast and fluorescence (**b**) sideview images of a microtissue before and after platelet-mediated contraction. Micropillar deflection ($\delta 1$ and $\delta 2$) can be easily detected after platelet adhesion and activation (postflow), which was used to calculate microclot-generated contractile force. Comparison of microtissue contractile force (**c**) and fluorescence intensity (**d**) before and after platelet flow. **e** Schematic diagram of microtissues before and after externally applied stretching. Mechanical stretching caused microtissue elongation (d2−d1), which was used to calculate tensile strain; the force needed to develop such elongation was reported by micropillar deflection as $F = k(\delta 1' + \delta 2')/2$ which was used to calculate the tensile stress. Microtissue stiffness was determined based on stress−strain relationship. **f** Comparison of tissue elongation between a microclot and a bare collagen microtissue under the same amount of substrate stretch. **g** Comparison of the stiffness between microclots and bare collagen microtissues. *$P < 0.001$, $n > 10$, each dot in box plot represents an independent experiment. All box plots with whiskers represent the data distribution based on five number summary (maximum, third quartile, median, first quartile, minimum). Statistical significance was determined by unpaired $t$ test with Welch's correction method. Scale bar is 200 μm

accompanied by significant increase in tissue-level fluorescence intensity (Fig. 2c, d), confirming that contractile force is mainly generated by aggregated platelets. In contrast, shear flow without platelet has negligible effect on measured contractile force (Supplementary Fig. 9).

During clot remodeling, progressive clot retraction is often accompanied by clot stiffening[2]. In the clotMAT system, microclot stiffness was measured by tensile testing, which was enabled by stretching the bottom silicone membrane (Fig. 2e, Supplementary Fig. 10a–c). Externally applied tensile force was reported by micropillar deflection as $\bar{F} = k\ (\delta 1' + \delta 2')/2$, and the microtissue tensile strain was determined based on image tracking of the displacement of fiducial markers in the microtissue before and after stretch (Supplementary Fig. 11). The stress and strain relationship derived from tensile testing

results was then used to calculate microclot stiffness (Supplementary Fig. 10d). Under the same amount of substrate stretch, the forced elongation of the microclot ($\delta 1'_{micro\_clot} + \delta 2'_{micro\_clot}$) was less than that of bare collagen microtissue ($\delta 1'_{collagen} + \delta 2'_{collagen}$) (Fig. 2f), indicating higher resistance to the stretch and thus higher stiffness of the microclots as compared to that of the bare collagen microtissues. The stiffness of the bare collagen matrix ($5.7 \pm 0.8$ kPa, 36 replicates) falls within the range that has been shown to be optimal for platelet adhesion and activation[15,16], and thus it allows the modeling of platelet adhesion on physiologically soft subenthothelial matrix and is a substantial improvement over collagen coating on rigid substrates in existing microfluidic devices (Fig. 2g). The stiffness of microclots ($18.0 \pm 2.8$ kPa, 34 replicates), formed under 30 min of citrated PRP flow at $500\,\mathrm{s}^{-1}$ shear rate, is much higher than

that of the bare collagen matrix and agrees well with the stiffness of laboratory-created thromboembolic analog[30]. Such difference in measured stiffness suggests that the clotMAT device is sensitive enough to detect the stiffness increase occurred during microclot formation and remodeling.

In addition to the standard sample preparation condition (citrated PRP), we also tested the utility of the clotMAT system under several other common preparation conditions, including the use of whole blood, recalcified PRP samples and different anticoagulants. Since red blood cells (RBCs) interfere with optical detection, we washed away unbounded RBCs at the end of 30 min of whole blood flow at $500 \, s^{-1}$ shear rate and measured the microclot properties (Supplementary Fig. 13). Here, we observed that the size of the microclot formed with whole blood is larger and the contraction force is higher as compared to the microclots formed with PRP. Recalcification of citrated PRP significantly increased platelet adhesion, leading to the formation of a large, loose clot that hung above the microtissue and blocked the channel after 10 min of flow. Due to the relatively short flow period, we did not detect significant difference in the contraction force between microclots formed under citrate PRP vs. recalcified PRP (Supplementary Fig. 14). We also compared the properties of microclots formed under different anticoagulants. PRP samples were prepared using either sodium citrate or D-phenylalanyl-L-prolyl-L-arginine chloromethyl ketone (70 μM PPACK, a thrombin inhibitor). Here, we observed more platelet binding and higher contractile force generation in the early stage (first 10 min) of flow experiment in the PPACK group compared to the citrate group (Supplementary Fig. 15). Even though platelets are robustly activated by collagen regardless of the anticoagulant (Supplementary Fig. 16), PPACK restricts thrombin-mediated fibrin formation (Supplementary Fig. 17). The lack of fibrin commonly resulted in platelet detachment/thromboembolism in the PPACK run and this is observed as a decrease in platelet fluorescence intensity and contractile force in the mid to late stage (10–30 min) of the flow experiment in the clotMAT system (Supplementary Fig. 15)[31]. Treatment of the PPACK group with thrombin receptor activator peptide 6 (TRAP 6) rescued the decrease in fluorescence intensity during mid to late stage of the flow experiment in the clotMAT system but had no effect on contractile force (Supplementary Fig. 15). Overall, while fibrin formed by locally released thrombin from platelets may contribute to clot formation, stabilization and part of the contraction, contractile forces generated by the activated platelets and transmitted via direct adhesive interactions between platelets and the collagen matrix may be the primarily driver of clot contraction. Additionally, these results demonstrate the ability of the clotMAT system to model and measure clot formation under various coagulation conditions.

**Microclot mechanics under various shear flow conditions**. As a demonstration of the integrated modeling and measurement capacity of the clotMAT system, we studied clot mechanics under various shear flow-mediated clotting conditions. Thus, fluid shear rate was varied from 3 to $1500 \, s^{-1}$, and microtissue contractile force, fluorescence intensity, stiffness and tissue volume were measured over 30 min of citrated PRP flow (Fig. 3a). For all the shear rates tested, we observed that microtissue contractile force and fluorescence intensity continuously increased over the 30-min period (Fig. 3b−d). However, microtissue width and topview area continuously decreased during the same period (Supplementary Fig. 18a, b). The changing rate of these parameters (i.e. the slope of time-dependent curves) increased with increased flow rate (Fig. 3b−d, Supplementary Fig. 18a, b), indicating a shear-rate-dependent alteration in microclot properties. Indeed,

comparison of results between different flow rates revealed that tissue fluorescence intensity, microclot contractile force and microclot stiffness increased with increased shear rate (Fig. 3e–g), but microclot thickness and volume decreased with increased flow rate (Fig. 3h, Supplementary Fig. 19). Together, the data indicate that increased shear rate promotes platelet adhesion, microclot contractile force generation, volume retraction and stiffening. Such shear-rate-dependent changes in microclot tissue mechanics may be due to the nature of shear-rate-dependent platelet adhesion.

**Platelet agonist and antagonist treatments on microclots**. We biochemically characterized the clotMAT system using a range of agonist/procoagulants and antagonists (Fig. 4a). First, in order to dissect the contributions of platelet number and the contractility of individual platelets to clot stiffness and contractility, we perfused 10 μM ADP or $5 \, U \, mL^{-1}$ thrombin over microclots that were previously formed under 30 min flow of citrated PRP at $500 \, s^{-1}$. Since platelets were absent in the agonists-containing perfusion buffer, the number of recruited platelets was comparable between this experiment and the standard flow condition. Consistent with this notion, neither ADP nor thrombin caused significant change in the microclot fluorescence intensity (Fig. 4b). However, both stimuli significantly increased microclot contractile force and stiffness, and decreased microclot volume (Fig. 4c–e, Supplementary Fig. 20), with the effects of ADP being greater compared to thrombin under the conditions tested. The stiffness of ADP-treated microclots (28.4 ± 6.1 kPa, 22 replicates) was 56% higher than that of untreated microclots and agrees well with the stiffness of fresh thrombi (26 ± 2.6 kPa) collected from carotid and cerebral arteries in stroke patients[30]. Since the above changes in the clot mechanical properties were not due to alterations in platelet number, these data confirm that individual platelet-generated contractile force, rather than simply platelet number, contribute to the measured changes in clot stiffness and volume. The more profound effect of ADP (MW 427 Da) compared to thrombin (MW 37 kDa) in this setting may be due to its smaller size which enables greater penetration into the platelet-dense clot, unlike thrombin which would be diffusion limited[32,33]. Indeed, when thrombin or ADP was added to citrated PRP just prior to flow, thrombin caused the formation of larger, fibrin-rich clots that blocked the flow channel within minutes (Supplementary Fig. 21).

To determine the molecular players regulating clotting in this model, we applied antagonists against the platelet glycoprotein receptor GpIbα epitope (mAb HIP1) that recognizes von Willerbrand factor (VWF), and also a GpIIb-IIIa receptor antagonist (Abciximab) at both venous ($500 \, s^{-1}$) and arterial ($1500 \, s^{-1}$) shear rates. Here, antagonist treatment significantly inhibited platelet binding to the collagen matrix and associated fluorescence intensity (Fig. 4f, g). Reduction in microclot contractile force and stiffness (Fig. 4h, i) and increase in microclot volume (Fig. 4j, Supplementary Fig. 20) were also observed under these treatments as compared to untreated PRP, indicating lack of functional platelet activity. Among the antagonists, inhibition of GpIIb-IIIa function by Abciximab blocked platelet recruitment, clot contraction and stiffening at all shear rates while the anti-GpIbα mAb (HIP1) was more effective only at the higher shear rate, $1500 \, s^{-1}$ (Fig. 4f–j). This is consistent with the known property of GpIbα−VWF interaction in the arterial shear regime[8,34]. Together, these observations are consistent with the notion that initial platelet recruitment is facilitated by GpIbα binding to VWF that is initially recruited onto the collagen matrix[35]. This is likely followed by the engagement of GpIIb-IIIa and other proteins that engage platelets

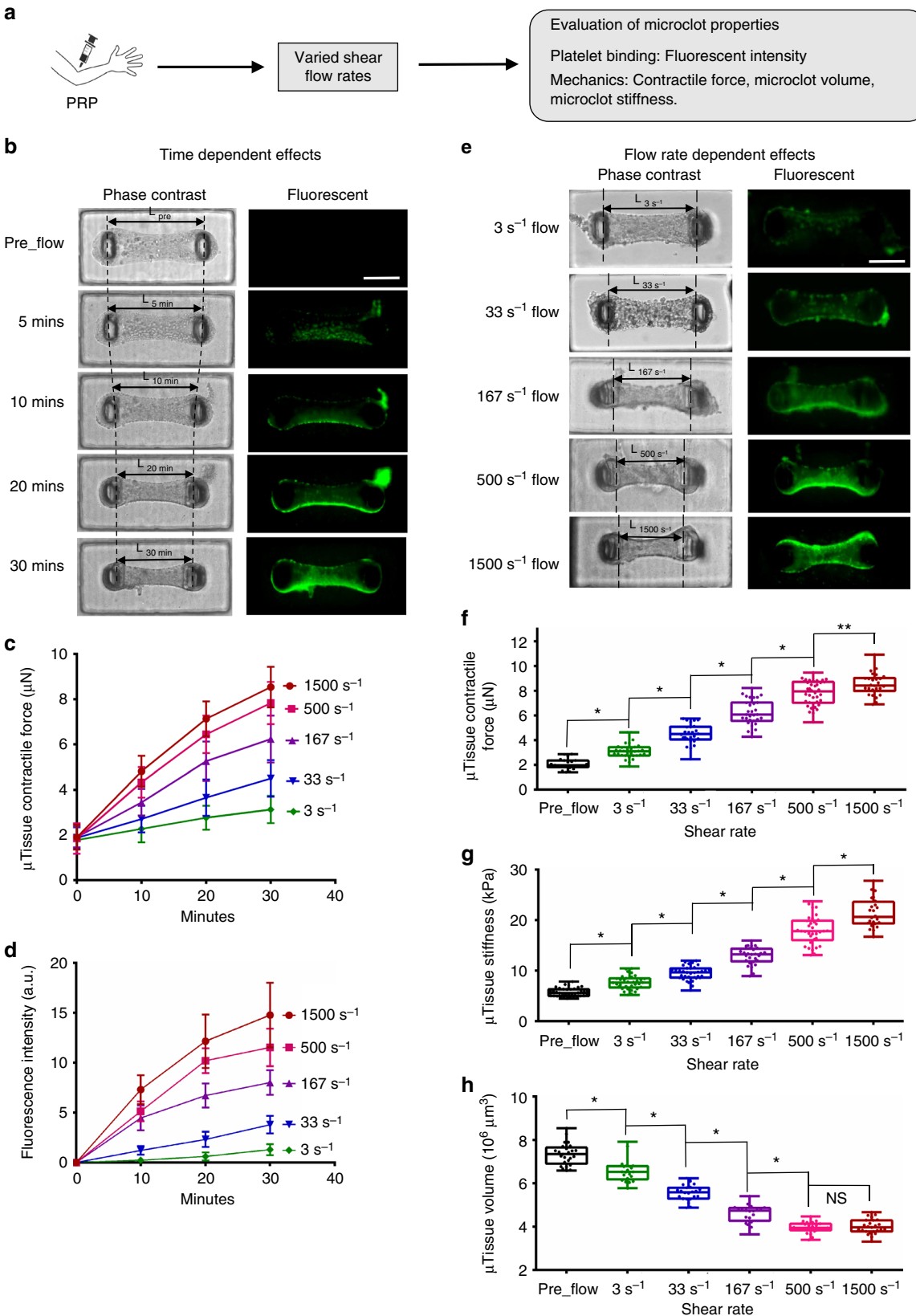

onto the extracellular matrix[36,37]. The colocalization of fibrin and platelet signals in confocal images suggests that platelet adhesion and fibrin formation occur over overlapping time scales (Supplementary Figs. 5 and 6).

In addition to platelet antagonists, we also used a myosin IIA inhibitor blebbistatin to simulate platelet myosin IIA impairment-induced clotting disorder, such as those seen in May-Hegglin anomaly, Fechtner, Epstein, and Sebastian

**Fig. 3** Microclot mechanics under various shear flow conditions. **a** Overview of the approach for evaluating microclot mechanics under various shear flow conditions. **b** Time-lapsed phase contrast and fluorescence images of a microclot formed under 167 s$^{-1}$ shear rate. The number of adhered platelets and microclot contraction gradually increased over a 30-min period, as demonstrated by gradually increased microtissue fluorescence intensity and decreased micropillar distance (L). Measured microclot contractile force (**c**), fluorescence intensity (**d**) over the 30-min time period for 3, 33, 167, 500 and 1500 s$^{-1}$ shear rates. **e** Phase contrast and fluorescence images of microclots formed after 30 min flow under different shear rates. Increased shear rate leads to increased level of platelet adhesion and microclot contraction, as demonstrated by increased microtissue fluorescence intensity and decreased micropillar distance (L) from shear rate 3 to 1500 s$^{-1}$. Measured contractile force (**f**), stiffness (**g**) and volume (**h**) of microclots formed at 30 min under different shear rates. Data are presented as mean ± standard deviation for (**c**), (**d**). *$P < 0.005$, **$P < 0.05$, $n > 10$, each dot in box plot represents an independent experiment. All box plots with whiskers represent the data distribution based on five number summary (maximum, third quartile, median, first quartile, minimum). Statistical significance was determined by one-way analysis of variance (ANOVA). Scale bar is 200 μm

syndromes[38]. Blebbistatin treatment in citrated PRP flow for 30 min at 500 s$^{-1}$ shear rate resulted in faint fluorescence (Fig. 4f, g), a significant reduction in microtissue contractile force and stiffness (Fig. 4h, j), and an increase in microtissue volume (Fig. 4j, Supplementary Fig. 20), as compared to untreated PRP sample. Such impairment on the microclot mechanical properties by blebbistatin is consistent with its effects on disrupting thrombus consolidation and stability reported in the previous study[39]. To examine the diagnostic potential of the clotMAT system, we tested the response of the clotMAT system to acetylsalicylic acid (ASA, aspirin) treatment. ASA treatment of citrated PRP resulted in significant reductions in microtissue fluorescence (Fig. 4f, g), microtissue contractile force, stiffness (Fig. 4h, i), and an increase in microtissue volume (Fig. 4j, Supplementary Fig. 20), as compared to untreated PRP sample, which is consistent with previous study[40]. Overall, these data demonstrate that the clotMAT mimics molecular aspects previously reported in classical flow chamber/microfluidics systems in collagen-based substrate assays[8]; only now we can also simultaneously measure platelet contractility and clot stiffness.

**VWD patient sample results in mechanically weak microclots.** We determined if the clotMAT system can be used for the diagnosis of bleeding disorders. To this end, we used VWD (von Willebrand Disease) type 2A patient plasma due to the prevalence of this disease and the role of VWF on platelet adhesion in this system (Fig. 5a). In these patients, enhanced ADAMTS13-mediated proteolysis results in deficiency in high molecular mass VWF multimers, lower plasma VWF concentrations and reduced shear-induced platelet activation (Supplementary Fig. 22). Flow experiments using reconstituted VWD PRP (rVWD PRP) for 30 min at 500 s$^{-1}$ resulted in faint fluorescence in microtissues, indicating significant reduction in platelet binding to collagen (Fig. 5b, c). Microtissue contractile force and stiffness were also lower and microtissue volume was higher in reconstituted VWD sample as compared to control samples including both healthy PRP and reconstituted healthy PRP (rHealthy PRP) (Fig. 5d−f). Together, these results demonstrate that the clotMAT device is sensitive enough to detect the change in clot mechanics associated with bleeding disorders.

**Effects of biochemicals and shear flow on clot stiffening.** The ability of the clotMAT system to report major mechanical parameters involved in clotting process enables us to perform integrated, quantitative analysis of these parameters. Using data collected from different clotting experiments performed under various flow and coagulation conditions, we performed regression analyses of the microclot contractile force against its stiffness and volume. We then compare the regression slopes between flow-mediated condition and biochemically induced coagulation conditions. Here microclot volume is used as a quantitative

measurement of the clot retraction level. For biochemically induced coagulation conditions (procoagulation factors, antagonists, VWD patient sample), we observed strong correlations between microclot stiffness and its contractile force ($R^2 = 0.97$, Fig. 6a, Supplementary Fig. 23) and between microclot volume and its contractile force ($R^2 = 0.93$, Fig. 6b). For all shear rates tested (3−1500 s$^{-1}$ shear rate), strong correlations between these mechanical parameters were also observed ($R^2 = 0.96$−0.97, Fig. 6c, d, Supplementary Fig. 23). Interestingly, no significant difference was found for the regression slopes between these two experimental groups, indicating these two types of stimulants independently yet equally strongly affect clot remodeling and stiffening.

To further dissect the biomechanical parameters involved in clot contraction and retraction, we performed finite element (FE) analysis of the microclot retraction process by representing the contractile platelet population using contractile 3D elements[27,41]. Due to the geometrical restriction by two micropillars at both ends, microclot contraction causes volume shrinkage and the development of necking in the middle section (Fig. 6e, Supplementary movie 3), mimicking the microclot retraction process observed in the experiment. Simulated decrease in microclot volume almost linearly correlates with increased contractile stress across all force ranges (Fig. 6f), recapitulating the experimental correlation between these two measurements, as shown in Fig. 6b, d. These results suggest that clot retraction is driven by the collective contractile force of the platelet population, which may be activated by biochemical treatments and shear flows.

**Discussion**
The mechanical properties of blood clots are critical to stem bleeding upon vessel injury, and may also contribute to embolism. Despite numerous studies on platelet mechanics[2,14,15], little is known about the origin of tissue-level clot mechanical properties[1,29]. Commonly used tissue mechanics models such as cell-laden bulk hydrogel models do not recapitulate the dynamic flow environment that is critical to clot formation and are low-throughout and time and resource intensive[42,43]. We described herein a medium-throughput clotMAT system with integrated flow and mechanical measurement capacity. Using a panel of agonist and antagonist used in classical vascular injury models, we demonstrate the ability of this system to integrate the biomolecular players and shear stresses that are known to be critical in vivo. The system recapitulates the evolution of clot morphology and mechanical properties under normal and abnormal clotting conditions. Since clot mechanical properties are widely used in the clinic as indicators for blood coagulation functions, the knowledge gained in this study may help to improve the diagnosis and treatment of coagulation disorders.

Initial platelet adhesion in the clotMAT system is dependent on collagen, VWF, and the platelet receptors GpIbα and GpIIbIIIa. In this regard, whereas blocking GpIIbIIIa inhibits platelet accumulation at all shear rates, GpIbα−VWF interactions

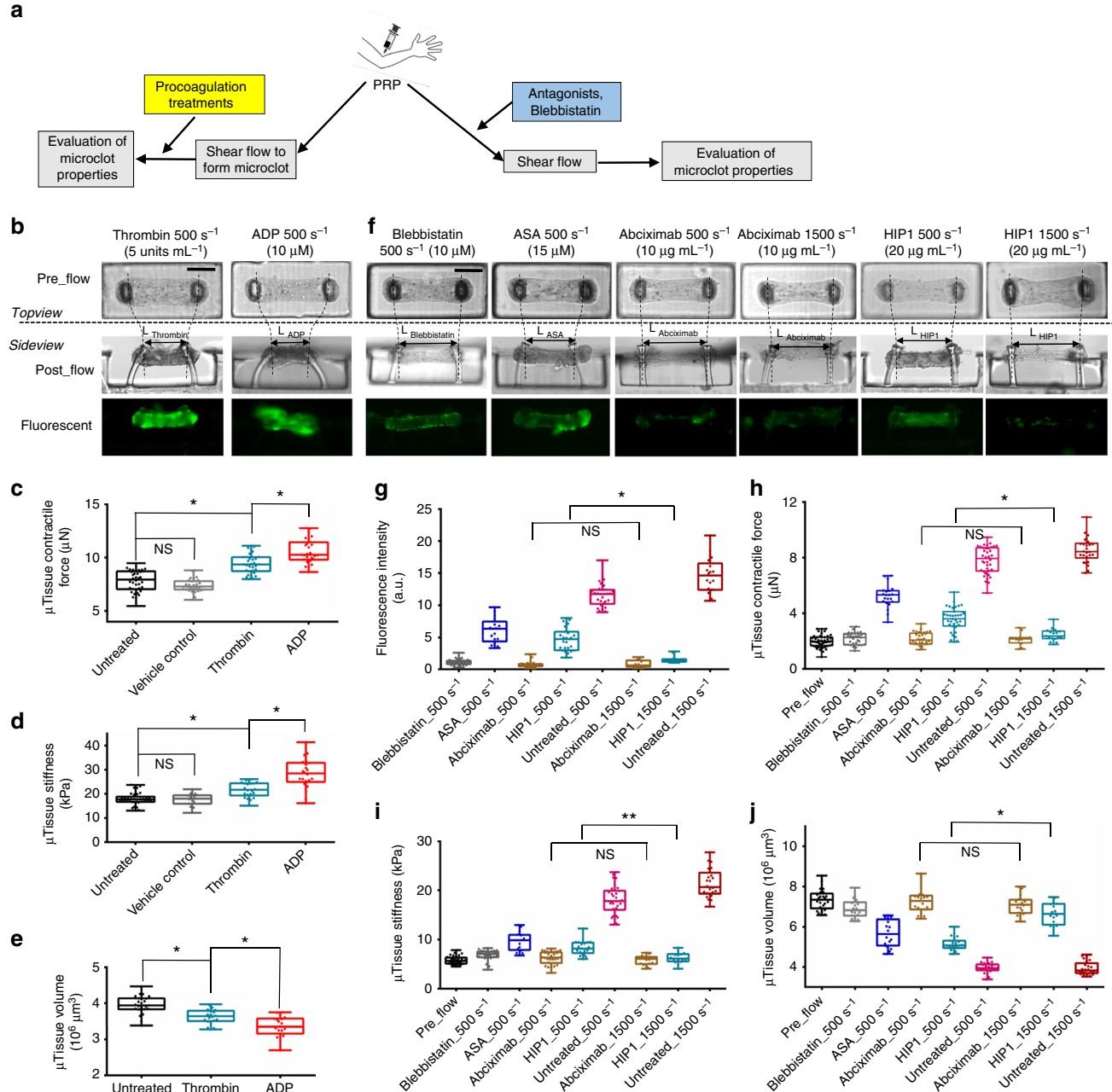

**Fig. 4** Microclot mechanics under platelet agonist and antagonist treatments. **a** Overview of the approaches. **b** Phase contrast and fluorescence images of microclots after 10 μM ADP or 5 U mL$^{-1}$ thrombin treatment under flow. These microclots were first formed over 30 min of 500 s$^{-1}$ platelet flow prior to agonist introduction for an additional 10 min. Measured contractile force (**c**), stiffness (**d**) and volume (**e**) of microclots after various procoagulation treatments. **f** Phase contrast and fluorescence images of microtissues formed after 30 min of 500 or 1500 s$^{-1}$ platelet flow treated with blebbistatin, acetylsalicylic acid (ASA), abciximab (anti GPIIb/IIIa) or HIP1 (anti-GpIbα) antibody. Measured fluorescence intensity (**g**), contractile force (**h**), stiffness (**i**) and volume (**j**) of microtissues formed under above anticoagulation treatments. **P $< 0.05$, *P $< 0.005$, $n > 10$, each dot in box plot represents an independent experiment. All box plots with whiskers represent the data distribution based on five number summary (maximum, third quartile, median, first quartile, minimum). Statistical significance was determined by one-way analysis of variance (ANOVA). Scale bar is 200 μm

are dominant modifiers of platelet adhesion at arterial shears (Fig. 4f–j). The observation that a fibrin-platelet shell covers the collagen substrate suggests that thrombin secreted locally at the coagulation site may contribute to fibrin formation (Supplementary Figs. 5–7). Consistent with this, fibrin deposition is only observed on the collagen substrate and not other sections of the flow channel (Fig. 1). Reduction in fibrin formation upon use of PPACK, a thrombin inhibitor, resulted in unstable platelet thrombi that embolized under flow (Supplementary Fig. 15).

Nevertheless, clot contraction was measured even upon use of PPACK, although it was ~2 μN lower compared to the case of citrated PRP. Thus, contractile forces in the system are likely primarily generated by the activated platelets and transmitted via platelet interaction with the collagen substrate, presumably via receptors GpVI and VLA-2[37]. Fibrin has an indirect role in clot retraction as it stabilizes the platelet thrombi and prevents embolization. Addition of thrombin or ADP to the flowing PRP results in higher amounts of fibrin formation and larger platelet

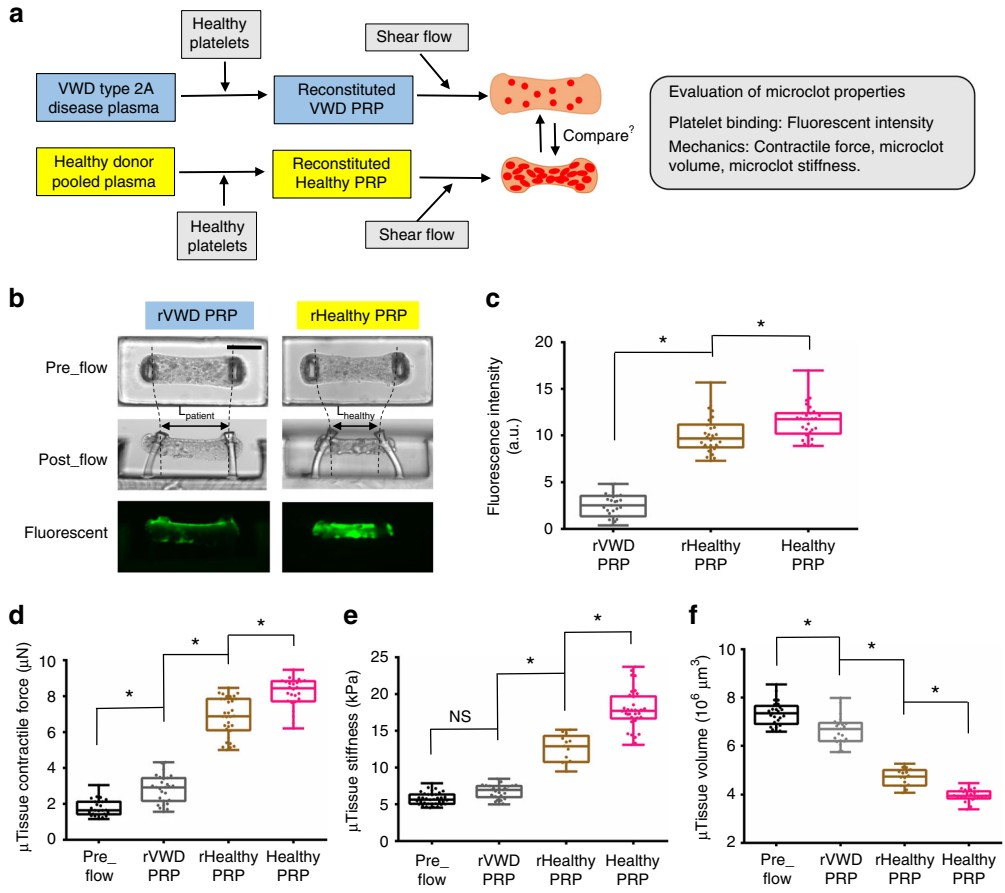

**Fig. 5** VWD patient sample results in mechanically weak microclots. **a** Overview of the strategy for comparing microclot mechanics between VWD sample and healthy donor sample. Healthy human washed platelets were added to VWD patient plasma and healthy donor plasma to form reconstituted VWD PRP (rVWD PRP) and reconstituted healthy PRP (rHealthy PRP). The mechanical properties of microclots formed using these reconstituted systems was compared with "Healthy PRP", normal PRP isolated from healthy volunteer. **b** Phase contrast and fluorescence images of microclots formed using VWD patient sample and healthy donor sample. Measured fluorescence intensity (**c**), contractile force (**d**), stiffness (**e**) and volume (**f**) of the above two microclot groups. *$P < 0.05$, $n > 10$, each dot in box plot represents an independent experiment. All box plots with whiskers represent the data distribution based on five number summary (maximum, third quartile, median, first quartile, minimum). Statistical significance was determined by one-way analysis of variance (ANOVA)

filled clots (Supplementary Fig. 21). Clot contraction and the impact of fibrin could not be evaluated during the course of this study, due to the massive platelet-fibrin clots that blocked the flow path. While tissue factor is absent in the current study, it is not possible to rule out the possibility that the contact pathway may also contribute to platelet deposition and clot mechanics in this system.

In clinical hematology, a major existing challenge is the diagnosis of patients whose standard laboratory test results are borderline but still have bleeding symptoms. Currently, there is no good functional assay that can be used to diagnose these patients. Thromboelastometry such as TEG and ROTEM are among the most commonly used point-of-care devices for diagnosing coagulation disorders, but it suffers from apparent limitations in diagnosing VWD and disorders of primary hemostasis[18,19]. In the current study, we showed that clotMAT system is able to detect impaired clot formation and stiffness of VWD patient samples. Such improvement over existing clinical tests suggests that the clotMAT system may have the potential to be used to improve the diagnosis of coagulation disorders, especially for patients whose clinical phenotype cannot be determined using existing clinical tests. Preliminary results in the current study show that the clotMAT system is able to report time-dependent increases in clot

mechanical properties based on the smooth ascending curves (Fig. 4b), which is qualitatively similar to the ascending curves representing clot stiffening in clinical thromboelastometry[18,19].

To further validate the diagnostic potential of the clotMAT system, we also tested its response to ASA since patients at risk to cardiovascular events commonly consume aspirin. The results showed that clotMAT is sensitive enough to detect the mild impairment of platelet function, clot formation and stiffness by ASA. This test, along with others performed using platelet antagonists (abciximab, anti-GpIbα) and blebbistatin, was performed using healthy blood samples collected from more than 20 healthy donors, yet the significant differences in the results were not masked by donor variation, which demonstrates the detection sensitivity of the system. Together, these data provide preliminary confirmation of the potential clinical utility of the clotMAT system. Additional comprehensive tests in a clinical setting are needed in the future to fully validate the clotMAT system. These tests may include reagents often used in TEG and ROTEM assays such as thromboplastin, heparinase and ellagic acid, different VWD/disease populations, and hyper reactive platelets from individuals suffering from cardiovascular diseases. It would be also interesting to compare the detection sensitivity of the clotMAT system directly with TEG and ROTEM. Finally, the

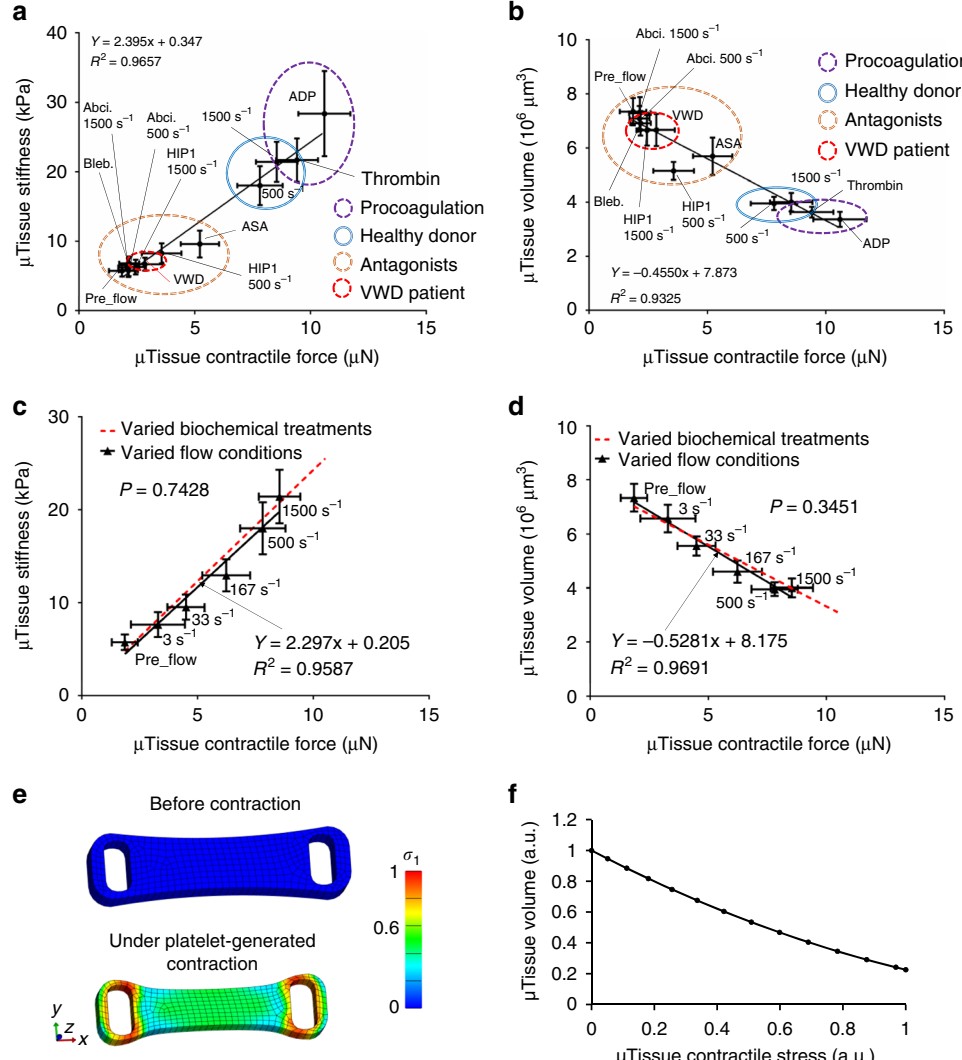

**Fig. 6** Effects of biochemical treatment and shear flow on clot stiffening. Strong correlation between microclot stiffness and microclot contractile force (**a**) and between microclot volume and microclot contractile force (**b**) exists for various coagulation conditions under biochemical treatments. Abciximab (Abci.), Blebbistatin (Bleb.), VWD (VWD patient sample), acetylsalicyclic acid (ASA), anti-GpIbα antibody (HIP1). **c**, **d** Comparison of linear regressions between biochemical treatment condition and shear flow condition. Strong correlation between microclot mechanical properties also exists under varied shear flow (solid, black line). No significant difference was found for the regression slopes between biochemical treatment condition and shear flow condition. The significant difference of slopes and intercepts for two linear regressions was measured using a method equivalent to analysis of covariance (ANCOVA). Correlation between groups was analyzed using Pearson correlation coefficients. **e** Finite element model of the microclot before and under platelet-generated contraction. Simulated effective stress contour is plotted over undeformed and deformed model geometry. Active contraction of platelet population is represented by the contractility of every single element. Due to the geometrical restriction by two micropillars at both ends, microclot contraction causes volume shrinkage and the development of necking in the middle section, mimicking the microclot retraction process observed in the experiment. Color scale indicates normalized effective stress level. **f** Simulated decrease in microclot volume almost linearly correlates with increased contractile stress, recapitulating the experimental correlation between these two measurements. Data are presented as mean ± standard deviation

development of the clotMAT system was enabled by the advancement in microfabrication technology; due to its low-cost, medium-throughput and the potential for scale-up, there exists a high potential to translate this technology into an improved point-of-care system for coagulation diagnosis[18].

The mechanics involved in tissue remodeling has often been studied using cell-laden bulk hydrogels comprised of matrix proteins[17,42,43]. However, due to the difficulty to incorporate flow condition and the inherent limitations in solute diffusion and experimental throughput, bulk hydrogel models are not suited for the study of clot mechanics. Previous studies using these models have shown that stiffening of the fibrin gels can be induced through either externally applied mechanical strain or myosin-

mediated cellular contraction that aligns and bundles fibrin fibers and compacts matrix meshwork[44,45]. However, the applicability of this strain stiffening theory under shear flow condition has not been examined. Here, through analyzing results from coagulation conditions regulated by either biochemical treatments or shear flows, we observed equally strong correlations in the microclot mechanical properties (Fig. 6a–d; Supplementary Fig. 19), indicating these two types of stimulants independently yet equally strongly affect clot remodeling and stiffening. This finding unveils the tissue remodeling/stiffening mechanism in a previously unexplored shear flow regime, and thus it significantly expands the boundary of previously established tissue remodeling theory that was based on bulk, static hydrogel models. Such integrated,

quantitative analysis of the tissue remodeling mechanism under flow condition is enabled by the combined tissue mechanics and shear flow capacity of the clotMAT system[44,46]. Adding cyclic stretching function to this system may further expand its capacity to detect viscoelastic property of the clots.

## Methods

**Healthy donor plasma sample preparation.** Human blood was obtained by venipuncture from healthy adult volunteers, following protocols approved by the UB Health Science Institutional Review Board (Buffalo, NY). Volunteers of either sex were used and studies were performed within 3 h of blood draw. Blood donors self-reported absence of coagulation defects and use of medication. Majority of the samples were prepared using sodium citrate (1:9) as anticoagulant with a few exceptions that were prepared using PPACK (70 μM, Cayman Chemicals). In some cases, PRP was recalcified by addition of 10 mM CaCl$_2$. 2 μM Prostaglandin E1 (PGE1, Cayman Chemicals) was added when washed platelets were prepared. PRP was obtained by centrifuging blood at $180 \times g$ for 15 min[47]. To obtain healthy plasma or platelet-poor plasma (PPP), the remaining blood was again spun at $1200 \times g$ for 12 min and the plasma supernatant was collected. Washed platelets were similarly obtained by further centrifuging PRP containing PGE1, at $1200 \times g$ for 12 min. In this case, the pellet was washed once using HEPES buffer (30 mM 4-(2-hydroxyethyl)-1-piperazineethanesulfonic acid, 110 mM NaCl, 10 mM KCl, 1 mM MgCl$_2$, 10 mM glucose, pH 7.4) containing 2 μM PGE1 before resuspension in HEPES buffer lacking PGE1 [35]. 1 μM BCECF (2′,7′-Bis-(2-Carboxyethyl)-5-(and-6)-Carboxyfluorescein, Acetoxymethyl Ester) (ThermoFisher Scientific) was added to PRP/washed platelets for 30 min at room temperature to stain platelets with green fluorescence.

**VWD type 2A patient plasma sample preparation.** VWD type 2 A patient plasma from a single donor was obtained from CoaChrom Diagnostica GmbH (Austria). 0.7% agarose gel electrophoresis was used to compare VWF multimer distribution in patient and healthy plasma[47]. ADAMTS13 activity was measured in terms of FRET ratio using the XS-VWF FRET substrate[48]. Briefly, this involved addition of citrated plasma VWF to 1 μM XS-VWF FRET for 1 h at room temperature. FRET ratio was then quantified using a Synergy 4 BioTek fluorescence plate reader, based on the ratio of XS-VWF emission intensities at 541/25 nm vs. 485/20 nm following excitation at 420/50 nm. A flow cytometer-bead sandwich assay determined VWF concentration[49]. Washed platelets (300,000 μL$^{-1}$) from healthy donor blood were mixed with either healthy plasma from normal donor to obtain reconstituted healthy PRP (rHealthy PRP) or VWD plasma to obtain reconstituted VWD PRP (rVWD PRP).

**Microtissue array device fabrication.** The microtissue array device was made by multilayer microlithography and soft-lithography techniques[20,21]. Briefly, multiple layers of SU-8 (bottom layer for the leg section and top layer for the head section) were successively deposited on the silicon wafer (University Wafer), exposed to UV light through transparency masks printed by laser plotting (CAD/Art Services Inc.), and baked and developed according to the manufacturer's protocols. Soft-lithography was then used to transfer the micro-patterns to polydimethylsiloxane (PDMS, Sylgard 184, DowCorning) molds made with 10:1 ratio of dimer to curing agent (Supplementary Fig. 1). The micropillar geometry was optimized through increasing micropillar height and reducing micropillar cross-sectional area to increase its force sensing sensitivity. The optimized micropillar dimensions are: width $(W) = 42$ μm, depth $(T) = 81$ μm, and height $(L) = 291$ μm (Supplementary Fig. 4). According to the cantilever bending theory, $F = k\delta = \frac{3EI}{L^3}\delta$, where $E$ is the Young's modulus of PDMS, $I$ is the moment of inertia, $L$ is the height of micropillar and $\delta$ is the deflection at the micropillar head, the spring constant $(k)$ of the micropillar is 120 nN μm$^{-1}$. The micropillar head was designed to stick out of the microwell so that when microtissues form on the heads, they can directly face and interact with the platelet flow in the microchannel to increase the platelet capture efficiency (Supplementary Fig. 4).

**Preparation of bare collagen microtissue array.** Human umbilical vein endothelial cells (HUVEC) were obtained from Lonza (C2519A, Walkersville, Maryland) and cultured in Endothelial Cell Growth Medium 2 (CC-22011, PromoCell). To perform microtissue seeding, unpolymerized collagen type I were neutralized and diluted to final concentrations of 3 mg mL$^{-1}$ (rat tail collagen I, #354236, Corning) and 20 μg mL$^{-1}$ (equine collagen, Chrono-log, Havertown, PA) and mixed with HUVECs (300,000 cells per device). This mixture was seeded into sterilized micropillar device through centrifugation, polymerized and maintained under standard culture condition for 2 days (Supplementary Fig. 1). By day 2, dog-bone-shaped microtissues were formed in the microwells. The microtissues were then treated with 0.25% trypsin for 20 min to allow most of the HUVECs to detach from the collagen matrix, resulting in a nearly bare collagen microtissue hung between the two micropillars (Supplementary Fig. 3).

**Fabrication and integration of the microfluidic channel.** The PDMS microchannels measuring 1189 μm in width and 498 μm in height were fabricated using standard photolithography and soft-lithography methods. The microchannels were aligned with individual rows of trypsinized microtissues under stereomicroscope, and the PDMS slab containing microchannels was bond with the PDMS substrate containing microtissue array through custom clamps (Supplementary Fig. 4). The platelet flow was created through syringe pump (NE-1000, New Era Pump System) withdraw at the outlet of the microfluidic channel, and the inlet of the microfluidic channel was connected to a platelet-containing reservoir (~300,000 platelets per μL, 8 mL total). Both the inlet and outlet of the microchannel were connected to the flow system through 18G needles. Flow rate was set between 0.01 and 4.5 mL min$^{-1}$ and the corresponding shear rate was 3~1500 s$^{-1}$, calculated by $\dot{\gamma}_x = \frac{6Q}{wh^2}$, where $Q$ is the volume flow rate, $w$ and $h$ are the width and the height of the channel, respectively. In the study of the effect of shear rate on microtissue formation, the experiments at each shear rate were performed on at least three donors with at least seven samples per donor.

**Microtissue contractile force measurement.** HUVEC-mediated microtissue formation involves the generation of contractile force that partially remains after the removal of the HUVECs by trypsinization. Such residual contractile force of $1.88 \pm 0.49$ μN (153 replicates) was recorded before platelet flow. During platelet flow, bright field images of the micropillars were taken every 10 min to monitor platelet-generated contractile forces in real time. Micropillar deflection was determined by the travel distance of the micropillar head relative to the bottom of its leg and was used to calculate the microtissue contractile force according to the cantilever bending theory $F = k\delta$, where $\delta$ is the averaged deflection $\delta = (\delta 1 + \delta 2)/2$ of the two micropillars and $k = 120$ nN μm$^{-1}$ is the spring constant of the micropillar (Fig. 2a and Supplementary Fig. 5). A Nikon Eclipse Ti-U inverted microscope with ×10 objective was used to image individual microtissue. During microclot contraction, each of the micropillars displaced around 50 μm during the 30 min PRP flow at 500 s$^{-1}$ shear rate, so the contraction rate of the microclot is approximately 1.7 μm min$^{-1}$, which is comparable to the value reported in the previous study[50].

**Microtissue stiffness measurement.** To enable mechanical stretching, microtissue array device was casted on a deformable silicone substrate (silicone sheeting, 0.01-inch NRV G/G 40D, SMI, Saginaw, MI) that is mounted on a custom-made uniaxial stretching frame (Supplementary Fig. 9). Silicone substrate was stretched to 150% strain along the longitudinal direction of the microtissues to apply uniaxial tension across the entire microtissue array. Images of the microtissue and micropillars before and after stretching were recorded and used to calculate the stress and strain. Externally applied stretch caused microtissue elongation ($d2–d1$), and the force needed to develop such elongation was reported by micropillar deflection as $F = k (\delta 1' + \delta 2')/2$. This force was divided by the cross-sectional area of microtissue to calculate the tensile stress (Fig. 2e). The microtissue tensile strain was determined based on microtissue elongation as $\varepsilon = (d2–d1)/d1$, by tracking the change in distance between two fiducial markers in the microtissue before and after stretch (Supplementary Fig. 11). Subsequently, the stress was divided by the strain to calculate the elastic modulus for the microtissue. The thickness ($t$) of the microtissue for cross-sectional area calculation was determined by confocal microscopy imaging (Supplementary Fig. 12). Due to the small thickness of the assembled clotMAT device (3−4 mm total), uniform stretching across the whole device was achieved without delamination between the layers.

**Platelet agonist and antagonist treatments.** Many assays in the current study were performed using PRP containing various platelet agonists/antagonists. These treatments include 10 μg mL$^{-1}$ Abciximab (anti-GpIIb/IIIa, Reopro, Eli Lilly), 10 μM blebbistatin (Sigma), 20 μg mL$^{-1}$ anti-CD42b antibody (#14-0429-80, HIP1 clone, ThermoFisher), and 15 μM acetylsalicylic acid (ASA, TCI America). Here, mAbs and ASA were incubated with citrated PRP for 30 min before flow and blebbistatin was incubated for 5 min before flow. In some cases, 15 μM TRAP-6 (#4017752, Bachem, Torrance, CA) was added to PRP prepared using PPACK just before flow. ADP (10 μM) and thrombin (5 U mL$^{-1}$) were diluted in Dulbecco's phosphate-buffered saline and infused at 500 s$^{-1}$ shear rate for 10 min over microclots previously formed under citrated PRP flow (Fig. 4b−e). In some cases, ADP (5 μM) and thrombin (2 U mL$^{-1}$) were directly added to citrate PRP before flow to study their effects on clot formation (Supplementary Fig. 21). For all the agonist and antagonist treatments, at least three donors with at least seven samples per donor were measured.

**Platelet activation studies.** PRP was isolated from fresh human blood using either 70 μM PPACK or 3.8% sodium citrate (1:9) as anticoagulant. In some cases, PRP was recalcified by addition of 10 mM CaCl$_2$. 50 μL PRP was incubated with a mixture of anti-human CD41/CD61 activation-specific mAb PAC-1 (Alexa647 labeled, #362805, Biolegend, San Diego, CA) and PE-labeled anti-CD62P mAb AC1.2 (#550561, BD, San Jose, CA) for 5 min at room temperature. This mixture was then stimulated with either 15 μM TRAP-6 (#4017752, Bachem, Torrance, CA), 5 U mL$^{-1}$ Thrombin (#T7009, Sigma, Burlington, MA), 20 μM ADP (Sigma, Burlington, MA) or 5 μgmL$^{-1}$ collagen (Chrono-Log, Haverton, PA). A portion of

the sample was diluted 200-fold at various times and analyzed using a BD Fortessa X-20 flow cytometer (Supplementary Fig. 16).

**Fibrin formation assay under static condition.** Alexa fluor-594-conjugated fibrinogen (Fb-594) was prepared by mixing 10 mgmL$^{-1}$ fibrinogen (#9001-32-5, Sigma) with 20-fold molar excess Alexa flour-594 (NHS ester, #1101-1, Fluoroprobes, Scottsdale, AZ) in PBS (pH = 8.0) for 1 h at room temperature. Following quenching of the reaction using 100 μL 1 M Tris-HCl (pH = 8.0) for 10 min, labeled protein was desalted into PBS using a Zeba Spin Column according to the manufacturer's instructions (7K MWCO, #89882, ThermoFisher).

To assay fibrin formation, 0.125 mg mL$^{-1}$ Fb-594 was added to PRP (20 μL volume). Following this, various stimuli were added: 15 μM TRAP-6, 5 U mL$^{-1}$ Thrombin, 20 μM ADP or 5 μg mL$^{-1}$ collagen. Such assays were performed using citrated PRP, recalcified citrated PRP and PPACK anticoagulated PRP under static condition. Following 10 min, 150 μL PBS containing 5 mM EDTA was added to stop the reaction. Fibrin clot was then immediately pelleted using centrifugation at 14,000 × g for 10 min, and Alexa 594 fluorescence in supernatant was measured using a plate reader in order to determine % Fb-594 incorporated into fibrin clot (Supplementary Fig. 17). Calibration curve was made by serial dilution of Fb-594.

**Shear-induced platelet activation (SIPAct).** Two microliters citrated healthy PRP was diluted 40-fold into either healthy human plasma (PPP) or VWD type 2A plasma. The mixture was sheared in a cone-plate viscometer at 9600 s$^{-1}$ [47]. Samples withdrawn at various times were incubated with Annexin-V FITC for 5 min at 37 °C in HEPES buffer containing 5 mM CaCl$_2$, prior to flow cytometry analysis. % Platelet activation is a measure of the fraction of platelets binding greater than baseline levels of Annexin-V (Supplementary Fig. 22).

**Immunofluorescence staining and imaging.** Microtissues were fixed using 4% (v/v) paraformaldehyde (PFA) (EMS, Hatfield, USA) for 10 min at room temperature, permeabilized by 0.1% (v/v) Triton X-100 for 5 min, and blocked with 3% BSA (Sigma) for 30 min at 37°C. Samples were incubated with collagen type I antibody (Millipore, AB755P, 1:300) overnight and labeled with Alexa Fluor™ 594 secondary antibody (ThermoFisher, A-11037, 1:400) for 1.5 h. Hoechst 33342 (Invitrogen) was used at 1:1000 ratio to label cell nuclei in the microtissue. Confocal images of the microtissue were taken using an Andor Technology DSD2 confocal unit coupled to an Olympus IX-81 motorized inverted microscope. Plan-Apochromat ×10 objective was used to record the stack with 1 μm optical slices for all channels. The stack of images was then processed using the Z stack tool in ImageJ (NIH) to obtain the projected 2D views. To characterize the microclot volume change after flow, topview area and thickness of the microtissue were analyzed based on bright field topview images and confocal stacks in ImageJ (Supplementary Fig. 12). To compare the signal intensity of platelet aggregation on microclots formed under various flow and coagulation conditions, images were taken by a Nikon Eclipse Ti-U inverted microscope equipped with ×10 air objective and Hamamatsu ORCA-Flash 4.0 LT CMOS camera under exact imaging conditions and processed in ImageJ. The video of platelet adhesion to the collagen microtissue was taken using the Nikon microscope and Flash 4.0 LT camera under fluorescence illumination at 1.25 fps frame rate for 4 min. A total of 300 images were recorded and processed in ImageJ.

**Finite element modeling of microclot retraction.** Finite element model of the microclot was constructed in FEBio[41]. Microclot geometry was discretized by 3D quadratic tetrahedral elements capable of large deformation. Microclot model is under active contraction and is restricted at the same time by two micropillars at both ends. The contractile platelet population was represented by the contractile elements whose constitutive material model consist of a neo-Hookean solid component and an isotropic contractile stress component[51]. The compressive neo-Hookean solid allows compaction of the microclot model under active contraction. Simulated effective stress contour was plotted over undeformed and deformed model geometry in the results. The change in microclot volume versus effective stress over the entire retraction process was plotted.

**Scanning electron microscopy.** The microtissues were fixed using 2% glutaraldehyde (233280250, ACROS organics) for 1.5 h, and dehydrated through a series of ethanol treatment at 15, 30, 50, 70, 90 and 100%. Finally, hexamethyldisilazane was added to the sample to prepare them for imaging on a HITACHI SU-70 SEM system.

**Statistics.** For all the shear flow conditions and all the agonist and antagonist treatment conditions, at least three donors with at least seven samples per donor were measured. Data are presented as mean ± standard deviation unless otherwise stated. Significance difference between dual comparison was verified by unpaired *t* test with Welch's correction method. Significance difference for multiple groups was determined by one-way analysis of variance (ANOVA). Correlation between groups was analyzed using Pearson correlation coefficients. The significant difference of slopes and intercepts for two linear regressions was measured using GraphPad Prism, using a method equivalent to analysis of covariance (ANCOVA).

**Reporting summary**. Further information on research design is available in the Nature Research Reporting Summary linked to this article.

## Data availability

All data supporting the findings of this study are available within the article and its supplementary information files or from the corresponding author upon reasonable request.

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

## Acknowledgements

Research reported in this study was supported in part by National Institutes of Health (NIH) under grants R01EB019411 (R.Z.), HL77258 and HL103411 (S.N.). The content is solely the responsibility of the authors and does not necessarily represent the official views of the National Institutes of Health. The authors would also like to acknowledge the funding support from the School of Engineering and Jacobs School of Medicine and Biomedical Sciences at the University at Buffalo.

## Author contributions

R.Z. and S.N. conceived the idea; Z.C., J.L., C.Z., I.H., X.Y., L.M., E.M., M.A. and R.Z. performed the experiments and analyzed the data; Z.C., S.J., S.N. and R.Z. wrote the manuscript.

## Additional information

**Competing interests:** The authors declare no competing interests.

