## [Peer Review File · Nature Communications]

Reviewers' comments:

Reviewer #1 (Remarks to the Author):

Authors solve a longstanding problem of measuring the mechanical properties of clots formed under hemodynamic conditions. A very novel microfluidic device allows creation of collagen structures supported by flexible microposts. Deposition of platelets and fibrin on this collagen, along with platelet retraction allows biological force generation and post deformation. In a very novel advance, the device allows the application of external force to stretch the clot assembly in order to determine passive mechanical clot properties. Microclot array elastometry has important implications for clinical phenotyping and basic research relevant to disease mechanisms and treatments.

1) Can the authors estimate the platelet density in their clots? In other flow assays and in laser injury, the platelets can occupy greater than 50 % of the clot volume. Is this density reached with PRP perfusion? Also, clots growth considerably faster with whole blood perfusion compared to PRP perfusion. Why was PRP used and not whole blood? Some experiments with whole blood perfusion would be more physiologic, particularly with triggering of the extrinsic pathway. Also, issues of citrate/recalcification/PGE1 may have subtle effects on platelet function and the investigators may wish to try different anticoagulation strategies.

2) The authors should emphasize the source of thrombin and fibrin in the clots, likely the contact pathway since there is no mention of tissue factor in the collagen. It was not clear if platelet accumulation and fibrin formation overlap temporally or occur sequentially. Does fibrin alter contractile force and clot stiffness? The fibrin in Fig. 1G-H is not particularly obvious and fluorescent staining would better detect the fibrin.

3) Authors should discuss the effects of viscoelasticity and creep which may be emeshed in their determination of the overall clot stiffness. Can the mechanical measurement be run in an oscillatory manner to give storage and loss moduli?

4) Authors should discuss and compare their results with Muthard et al. ATVB (2012;32:2938). who measured contraction rates and % strain due to contraction of clots formed under flow on deformable collagen.

Reviewer #2 (Remarks to the Author):

The mechanical properties of blood clots are critical for their hemostatic function and are influenced by a range of hemostatic components. These include platelet reactivity, adhesive function, contractility and fibrin mechanical properties and architecture. Clinically, several methods are used to test the mechanical properties of blood clots, including TEG and ROTEM. Recently, various microfluidic devices have been developed to assess the contractile force and stiffness of blood clots after their formation on immobilised matrices e.g. Myers et al . Nat Mater. 2017; 16(2):230-5. However, there are a paucity of clinical diagnostic assays that consider the influence of blood flow on clot mechanical properties.

The authors of this study have developed a new microfluidic device called clotMAT that models in vivo blood clot formation and allows quantitative measurement of 'contractile force' and 'stiffness' of forming blood clots under shear conditions. The device contains multiple paired micropillars, with each paired pillar heads joint by type1 collagen microtissue and pillar legs immobilized on to stretchable membranes. Only the pillar heads (microtissues) are in contact with flowing platelets and support platelet adhesion and aggregation. When platelet aggregates form and contract, the paired pillars bend towards to each other with the extent of bending reflecting the 'contractile force'. Concurrently, the paired pillar legs stretch the flexible membrane in an outward direction, with the extent of stretching reflecting the 'clot stiffness'.

Using this device, the authors demonstrated that perfusing platelet rich plasma at increasing shear

rates (3-500s⁻¹) led to an increase in contractile force and stiffness, and a corresponding reduction in aggregate size. Perfusing activators of platelets enhanced these responses, while inhibiting platelet adhesion, aggregation or contractility impaired these mechanical properties. Importantly, platelets resuspended in Type IIA vWD plasma (reduced vWf level and vWF multimer size due to enhanced ADAMTS 13 activity) displayed a reduced contractility and stiffness, with platelet aggregate size remaining unchanged. The authors' have concluded that both biochemical activation of platelets and shear greatly contribute to clot mechanical properties. The authors suggest that the device can mimic in vivo blood clot formation and predict the clot mechanical properties by accurately measuring force, stiffness and size of blood clots. The device therefore may represent a useful research and potential diagnostic tool to detect various bleeding disorders.

General comments

The device is new and innovatively designed, especially with the use of collagen microtissues. Although the authors have previously reported this technique related to microtissues (Zhao et al, *Biomaterials*, 2014, 35:5056), this device nevertheless represents an important technical advance in modelling clot formation. Its ability to simultaneously and quantitatively assess clot contractile force, stiffness and size under flow conditions is also technically highly innovative, despite that force measurement per se has been previously reported (Liang et al, *Lab on a Chip*, 2010, 10-991). In the end, all parameters measured under 11 conditions by this device i.e. stiffness, volume and contractile force all beautifully fit by one unified model (Figs. 6 and S13), which is quite impressive if well validated.

Overall, the studies are well designed and the manuscript and figures are well presented. However, there are numerous concerns that need to be resolved, including;

1) It is unclear why citrated PRP (red cell free), rather than whole blood was used to detect clot mechanical properties, as the absence of red cells would significantly affect platelet delivery/adhesion under flow. Furthermore, in clots formed from whole blood, the extent of clot contraction is limited by the presence of red blood cells (Tutwiler, *Biophys J* 2017). It would have been physiologically more relevant to have included whole blood in the clotMAT to determine clot elastometry. Also, if the authors intend their clotMAT for diagnostics, it would be useful to compare their elastometry with current applied devices, e.g. ROTEM.

2) The authors assessed the clot mechanical properties within a narrow shear range (3-500s⁻¹), mainly at venous rates, thus it is unclear if the device is also suitable for arterial or pathological shear rates.

3) Even though there is novelty in the design of the collagen matrix, the measurements of clot contractile force and stiffness have previously been studied with post force sensors for platelets (Liang, *Lab Chip*, 2010) or whole blood (Judith, *Lab Chip*, 2015).

4) It is not clear how consistent the collagen content is between individual microtissues within a clotMAT and between clotMATs, how dynamic platelet aggregates are on the microtissue, and how reproducible the assay is.

5) The diagnostic potential of clotMATs has not been adequately demonstrated, as only one type of vWf D type IIA plasma was tested, and this was tested at a suboptimal low shear rate (500s⁻¹). The diagnostic potential may be enhanced if this device can also detect the effects of anti-platelet drugs (eg aspirin), or hyper reactive platelets (eg platelets from patients with cardiovascular disease) or thrombocytopenic blood.

Specific comments:

6) Figure 1, it would be very helpful to illustrate all dimensions for all the features within a clotMAT: micropillars, microclots, top PDMS channel.

It would be also useful and important to describe platelet aggregate formation in a more detailed manner, since "submillimeter-size or thrombi (100um)" were formed as stated in Results page 7,

such size thrombi would likely disturb local flow and thus influence ongoing platelet adhesion, especially if the height of top PDMS channel is limited. How dynamic are the platelet aggregates? How reproducible is platelet aggregate formation between pillars, between donors, between clotMATs?

Furthermore, taking advantage of HUVEC contraction force, the authors used an innovative approach to pre-stress the microtissue by seeding the HUVEC then removing them with trypsinization. It seems that 1800 nN (as described in the Method) is critical for clotMATs improved performance over the bare-collagen setup and presumably other existing micro-fluidic devices. The authors need to provide the variability and range of such pre-stressing levels and the explain their effect on the clotting process.

Fig 1D, the collagen content appeared even in individual microtissues. Can the collagen content be quantitated for individual microtissues to ensure a consistent level of collagen, as it will greatly influence platelet aggregation (extent and size), thus force and stiffness?

Fig 1F-G, since anticoagulated PRP is used, why do the authors see fibrin mesh in the microclots?

7) Figure 2, the stretching membrane allows the detection of stiffness. Does it undermine contractile force measurement (pillar bending) (page 8)?

8) Figure 3 shows a shear-dependent change in contractile force, stiffness and platelet aggregate size. These findings validate the sensitivity and accuracy of the detection systems using clotMATs, however it does not support (at least not fully) the shear contribution to the mechanical properties, since these changes are more likely intrinsic, or greatly influenced by the number of platelets adherent/delivered at the indicated time and shear rates.

9) Figure 4 is interesting that ADP (10uM) had a greater impact than thrombin (5 U/ml) on clot mechanical properties. Any explanation? Are platelets on microclots fully activated? With such high concentration thrombin, the platelets are likely to become procoagulant and lose their adhesive and contractile function. Alternatively, the anticoagulant remaining in the PRP may diminish the platelet-activating effects of thrombin? If this is case, the thrombin receptor activation peptide would be preferable.

Fig 4B and 4F need a control. The impact of AK2 in Fig F-J seemed mild, in contrast to the importance of VWF (shown in Fig 5). Is it due to low shear rates used (500s-1), and platelet adhesion is not fully GPIb-VWF dependent at 500s-1. High shear rates should be tested using AK2 treated platelets.

The authors should re-evaluate the relative impacts of anti-GPIIb/IIIa, anti-integrin and anti-myosin treatments on platelet aggregation. Under the shear conditions employed, would anti-GPIIb/IIIa significantly reduce platelets recruitment? Blebbistatin certainly shouldn't. To my surprise, the impact of these treatments on this system is opposite to what would be expected. AK2 treatment had more platelet fluorescence than the other two; while blebbistatin had a profound effect on platelet aggregate size (4G,J), rather than selectively on contractile force and stiffness. What is the explanation?

10) Figure 5, it is unclear how the experiments were performed based on information provided in the figure legends and methods. It is thus hard to follow the results. Do the authors use normal platelets resuspended in vWD or normal plasma prior to perfusion through microMATs? If so, what do the "Washed platelet controls" mean in 5D, E, F?

Does the reconstituted PRP in vWD plasma show defects in ristocetin-induced platelet aggregation?

Were the vWD type IIA plasma from a single patient, or pooled? If pooled, how many patients were included?

11) Since anti-coagulated PRP is used, how much fibrin is generated in the assay? This is critical,

as fibrin is a major component of clot stiffness/contraction in vivo.

12) At the highest shear rate (500s^{-1}) used in this study, the platelet aggregate size kept increasing over a 30 min perfusion period. Is this relevant to physiological haemostatic responses?

13) The increase in clot contractile force most likely comes from the increased number of platelets adhered to collagen with increasing shear as shown in Fig 2C. However, the tissue stiffness with increasing shear must take into account the stiffening behaviour of fibrin in response to shear (Litvinov Matrix Biol, 2016). Please comment.

Other minor comments:

1. It would be useful to show a low magnification fluorescence image revealing platelet aggregates on multiple pillar heads to demonstrate reproducibility within a device, since the platelet deposition seemed heterogeneous on collagen microtissue (Fig 2B, Fig 3B,E, Fig 5B).

2. It is very impressive for the authors' team to make micro-pillar system for measuring clot retraction. Based on Fig.S4, it seems the aspect ratio of the micro-pillars are $>7:1$ (height:width). With such high aspect ratios, the pillars are very difficult to preserve and remain functional if damaged. It will be useful to provide more insight into how these technical challenges have been overcome.

3. The measurement of clot stiffness by membrane stretching is important. The information on how mechanical force is loaded and the force loading data are missing.

4. Do the authors have any insights whether this device is suitable for mouse blood studies?

5. Page 10, line 2. It mentions healthy donor's microclot measured in the paper is comparable to thrombi collected from stroke patients (26 kPa). This is potentially interesting, but how is this related?

6. The dimension of clotMats need to be more obviously illustrated (pillar leg height: 100um; pillar heads, collagen microtissue: 100um, top PDMS channel dimension).

We received reviewer comments on our manuscript, and we appreciate the time and effort each of the reviewers have dedicated to providing insightful feedback on our manuscript. In order to address the Reviewer comments, we have performed a series of new experiments that appear in main figures including Fig. 1, 3, 4, 5 and 6 and Supplementary figures including Fig. 2, 3, 5 – 7 and 10 – 23. This letter details our point-by-point response to reviewer comments and outlines the changes made to the manuscript. The changes made are also highlighted in both the main manuscript and supplemental material.

We reproduce reviewer comments in italics below, followed by our response.

Reviewer 1

Authors solve a longstanding problem of measuring the mechanical properties of clots formed under hemodynamic conditions. A very novel microfluidic device allows creation of collagen structures supported by flexible microposts. Deposition of platelets and fibrin on this collagen, along with platelet retraction allows biological force generation and post deformation. In a very novel advance, the device allows the application of external force to stretch the clot assembly in order to determine passive mechanical clot properties. Microclot array elastometry has important implications for clinical phenotyping and basic research relevant to disease mechanisms and treatments.

1) Can the authors estimate the platelet density in their clots? In other flow assays and in laser injury, the platelets can occupy greater than 50 % of the clot volume. Is this density reached with PRP perfusion? Also, clots growth considerably faster with whole blood perfusion compared to PRP perfusion. Why was PRP used and not whole blood? Some experiments with whole blood perfusion would be more physiologic, particularly with triggering of the extrinsic pathway. Also, issues of citrate/recalcification/PGE1 may have subtle effects on platelet function and the investigators may wish to try different anticoagulation strategies.

We thank the reviewer for the insightful comments. We have performed new SEM imaging (Supplementary Fig. 3, 7) and confocal fluorescence microscopy (Supplementary Fig. 5, 6) to characterize the structural composition of our microtissues and included new results in the main text and supplementary figures. These data show that the microtissue consists of a dog bone-shaped collagen core covered by a thick clot shell made of platelets and fibrin. Confocal reconstructed cross-sectional view of the microtissue shows that the clot shell thickness ($28.2 \pm 7.6 \mu\text{m}$) is approximately 78% the thickness of the collagen core ($36 \mu\text{m}$) (pg. 7 in Results section in main text, Supplementary Fig. 5, 6). This clot layer thickness is consistent with previously reported thickness of the clot formed at the vessel injury site in a mouse model¹. SEM and confocal images show that platelets uniformly distribute in the clot shell and occupy approximately $43.5\% \pm 11.6\%$ of its volume, which is also consistent with previously reported values¹. This density was achieved under citrated PRP perfusion.

We used PRP instead of whole blood in our previous experiments because red blood cells (RBCs) interfere with continuous optical detection, thus limiting our in situ measurements of microclot mechanical properties and geometry. Nevertheless, end-point studies are feasible with whole blood. To demonstrate the capacity of our system to work with whole blood, we have performed new flow experiments using whole blood and measured the microclot properties at the end of the flow after washing away unbounded RBCs (Supplementary Fig. 13). Here, we observed larger microclot formation and higher contraction force generation using whole blood as compared to PRP samples. These new data are included in pg. 9 in the Results section of the main text.

To understand the effects of different anticoagulants on system performance, we performed new experiments using PPACK (D-Phenylalanyl-prolyl-arginyl Chloromethyl Ketone, thrombin antagonist). Here, we observed more platelet binding and higher contractile force generation in the early stage (first 10 mins) of flow experiments in PPACK group vs. citrate group (Supplementary Fig. 15). Fluorescence intensity plateaued and then decreased and contractile force also decreased in the mid to late stages (10 – 30 mins) of the flow experiment for the PPACK group, suggesting platelet detachment and embolization¹. The weak strength of the clot may be due to the lack of fibrin mediated clot stabilization, as a result of the use of PPACK, since fibrin formation by thrombin is inhibited in PPACK (Supplementary Fig. 17). Treatment of the PPACK group with thrombin receptor activator peptide 6 (TRAP 6) rescued the decrease in fluorescence intensity during the mid to late stages of the flow experiment but had negligible effect on contractile force (Supplementary Fig. 15). This is consistent with cytometry observations that TRAP-6 is a potent activator of platelets even when PPACK is used as anti-coagulant, as it supports α -granule secretion (measured using P-selectin) and GpIIb-IIIa activation (detected based on PAC-1 binding) (Supplementary Fig. 16). Thus, its addition supports platelet attachment. Together, these results suggest that a balance between platelet functional activation and fibrin formation may control thrombus formation and contractility in the clotMAT system. Additionally, the data demonstrate the suitability of the clotMAT system to model and measure clot formation under different anticoagulation conditions, although citrated plasma appears to be the most suitable reagent. These new results have been included in pg. 9-10 in the Results section of the main text.

The majority of the experiments presented in this study were performed using citrated PRP without recalcification. As a comparison, we performed new experiments with recalcification, both using the clotMAT assay and flow cytometry. In the clotMAT assay, we observed that recalcification of citrated PRP significantly increased platelet adhesion, leading to the formation of a large, loose clot that was above the microtissue and that blocked the channel after 10 mins of flow (Supplementary Fig. 14). Calcification did not affect platelet activation markedly at the end point (Supplementary Fig. 16), but it increased fibrin formation (Supplementary Fig. 17) and it may also affect the kinetics of platelet activation. These new results have been included in the pg. 9-10 of Results section of the main text.

With respect to Prostaglandin E1 (PGE1), indeed this was used during washed platelet preparation. Here, we resuspended normal washed platelets in healthy donor plasma to make “reconstituted Healthy PRP (rHealthy PRP)” (Fig. 5 C-F). Upon application in the clotMAT assay, we only observed a minor reduction in platelet adhesion and contractile force generation upon use of ‘rHealthy PRP’ compared to ‘Healthy PRP’ prepared by standard one-step centrifugation in the absence of PGE1. Such minor decrease in platelet function would be expected following washed platelet preparation. We have relabeled these conditions in main Fig. 5 and pg. 21 in the Method section, in order to provide greater clarity in the manuscript.

2) The authors should emphasize the source of thrombin and fibrin in the clots, likely the contact pathway since there is no mention of tissue factor in the collagen. It was not clear if platelet accumulation and fibrin formation overlap temporally or occur sequentially. Does fibrin alter contractile force and clot stiffness? The fibrin in Fig. 1G-H is not particularly obvious and fluorescent staining would better detect the fibrin.

Thrombin in this system likely comes from platelets that are activated upon engaging primarily GPVI, and also VLA-2 on the collagen substrate. As the reviewer suggests, the contact pathway may also

contribute to thrombin secretion and fibrin formation. Such thrombin may remain bound to fibrin, thus enhancing local protease concentrations². This is evident based on the new confocal microscopy studies where fluorescence-labeled fibrinogen was observed to be deposited in the platelet-fibrin clot (Supplementary Fig. 5, 6). These platelet-fibrin deposits form on the outer surface of the collagen matrix to form a thick shell of clot ($28.2 \pm 7.6 \mu\text{m}$ thickness). Similar size estimates were also made using SEM (Supplementary Fig. 7). Indeed, we did not add lipidated tissue factor (TF) to the collagen substrate in this study, but we note in the revised Discussion that such studies could be performed in the future (Pg. 18, main text). Indeed, inclusion of CTI (corn trypsin inhibitor) and TF may enable the testing of this integrated device to study contact and tissue factor dependent processes, in addition to VWF and platelet related defects.

With respect to the temporal evolution, platelet deposition in the flow channel is seen as early as 5 min. (Fig. 3B) and fibrin deposition is observed by 10min., with the two signals co-localizing in the confocal images (Supplementary Fig. 6). These data suggest that platelet deposition and fibrin formation may occur on similar timescales, perhaps in a coordinated manner.

With regard to the contribution of fibrin to contraction and stiffness, we performed the clotMAT assay when using PPACK, conditions where fibrin deposition would be reduced (Supplementary Fig 15). In this instance, we observed significant contractile forces. In recalcified PRP experiment (Supplementary Fig. 14), we observed the formation of a large, fibrin-rich clot but the measured contractile force for not higher than the non-recalcified sample in the time-scale of the study. Together, these data suggest that while fibrin may contribute some part of the contractile force, a portion of the mechanical stress is likely also transmitted to the micropillars due to the direct interaction between the platelets and the collagen matrix. This is consistent with our previous study showing that ECM component contributes to the tissue stiffness, in other systems^{3,4}. The above points raised by the reviewer are addressed on pg. 7, 9-10 and 16-17 of the revised manuscript.

3) Authors should discuss the effects of viscoelasticity and creep which may be emeshed in their determination of the overall clot stiffness. Can the mechanical measurement be run in an oscillatory manner to give storage and loss moduli?

In our clot stiffness measurements, we performed the tensile tests in a “quasi-static” manner, ie. within short time period (several seconds), to ensure an elastic response of the microclot material. Our previous publications^{3,4} and a recent publication⁵ from author’s previous group showed that microtissue’s response to stretching is predominantly elastic within this time frame. At longer time frame, the effect of viscoelasticity may become apparent and may affect the measurement of the microclot properties. Our current stretching system can be coupled to a vacuum system to enable cyclic stretching that is needed for dynamic modulus (storage and loss moduli) measurements. We thank the reviewer for these suggestions and we have included these points as future work in pg. 19 of the discussion section.

4) Authors should discuss and compare their results with Muthard et al. ATVB (2012;32:2938). who measured contraction rates and % strain due to contraction of clots formed under flow on deformable collagen.

In the reference paper from Muthard et al.⁶, the contraction rate for the upstream clot was

measured to be around 1.5 $\mu\text{m}/\text{min}$ under 1130 s^{-1} shear rate. During microclot contraction in our system, each of the micropillars displaced around 50 μm during the 30 min PRP flow at 500 s^{-1} shear rate, so the contraction rate of the microclot is approximately 1.7 $\mu\text{m}/\text{min}$, which is comparable to the value in previous study. We have added this comparison in the Method section under Microtissue Contractile Force Measurement section pg. 23.

Reviewer #2 (Remarks to the Author):
The mechanical properties of blood clots are critical for their hemostatic function and are influenced by a range of hemostatic components. These include platelet reactivity, adhesive function, contractility and fibrin mechanical properties and architecture. Clinically, several methods are used to test the mechanical properties of blood clots, including TEG and ROTEM. Recently, various microfluidic devices have been developed to assess the contractile force and stiffness of blood clots after their formation on immobilised matrices e.g. Myers et al . Nat Mater. 2017;16(2):230-5. However, there are a paucity of clinical diagnostic assays that consider the influence of blood flow on clot mechanical properties. The authors of this study have developed a new microfluidic device called clotMAT that models in vivo blood clot formation and allows quantitative measurement of 'contractile force' and 'stiffness' of forming blood clots under shear conditions. The device contains multiple paired micropillars, with each paired pillar heads joint by type1 collagen microtissue and pillar legs immobilized on to stretchable membranes. Only the pillar heads (microtissues) are in contact with flowing platelets and support platelet adhesion and aggregation. When platelet aggregates form and contract, the paired pillars bend towards to each other with the extent of bending reflecting the 'contractile force'. Concurrently, the paired pillar legs stretch the flexible membrane in an outward direction, with the extent of stretching reflecting the 'clot stiffness'. Using this device, the authors demonstrated that perfusing platelet rich plasma at increasing shear rates (3-500s⁻¹) led to an increase in contractile force and stiffness, and a corresponding reduction in aggregate size. Perfusing activators of platelets enhanced these responses, while inhibiting platelet adhesion, aggregation or contractility impaired these mechanical properties. Importantly, platelets resuspended in Type IIA vWD plasma (reduced vWf level and vWF multimer size due to enhanced ADAMTS 13 activity) displayed a reduced contractility and stiffness, with platelet aggregate size remaining unchanged. The authors' have concluded that both biochemical activation of platelets and shear greatly contribute to clot mechanical properties. The authors suggest that the device can mimic in vivo blood clot formation and predict the clot mechanical properties by accurately measuring force, stiffness and size of blood clots. The device therefore may represent a useful research and potential diagnostic tool to detect various bleeding disorders.

General comments
The device is new and innovatively designed, especially with the use of collagen microtissues. Although the authors have previously reported this technique related to microtissues (Zhao et al, Biomaterials, 2014, 35:5056), this device nevertheless represents an important technical advance in modelling clot formation. Its ability to simultaneously and quantitatively assess clot contractile force, stiffness and size under flow conditions is also technically highly innovative, despite that force measurement per se has been previously reported (Liang et al, Lab on a Chip, 2010, 10-991). In the end, all parameters measured under 11 conditions by this device i.e. stiffness, volume and contractile force all beautifully fit by one unified model (Figs. 6 and S13), which is quite impressive if well validated. Overall, the studies are well designed and the manuscript and figures are well presented. However, there are numerous concerns that need to be resolved, including;

1) *It is unclear why citrated PRP (red cell free), rather than whole blood was used to detect clot mechanical properties, as the absence of red cells would significantly affect platelet delivery/adhesion under flow. Furthermore, in clots formed from whole blood, the extent of clot contraction is limited by the presence of red blood cells (Tutwiler, Biophys J 2017). It would have been physiologically more relevant to have included whole blood in the clotMAT to determine clot elastometry. Also, if the authors intend their clotMAT for diagnostics, it would be useful to compare their elastometry with current applied devices, e.g. ROTEM.*

We thank the reviewer for the encouragement and critical comments. A similar question has been answered for Reviewer #1, but we are happy to answer it here again. We used PRP instead of whole blood in our previous experiments because red blood cells (RBCs) interfere with optical detection, thus limiting our in situ measurements of the microclot mechanical properties and geometry. This may be remedied later by fluorescently tagging the micropillars. Nevertheless, to demonstrate the capacity of our system to work with whole blood, we have performed new flow experiments using whole blood and measured the microclot properties at the end of the flow after washing away unbound RBCs (Supplementary Fig. 13). Here, we observed larger microclot formation and higher contraction force generation using whole citrated blood as compared to citrated PRP. These new results have been included in pg. 9 in the Results section of the main text.

With regard to direct comparison with ROTEM, we are unable to do this as part of this revision since we do not have access to this instrument in our laboratories. This is, however, a valuable suggestion from the Reviewer and part of our future research plans as stated in pg. 18 in Discussion section.

2) *The authors assessed the clot mechanical properties within a narrow shear range (3-500s⁻¹), mainly at venous rates, thus it is unclear if the device is also suitable for arterial or pathological shear rates.*

As suggested by the Reviewer, we have performed new PRP flow experiments at 1500 s⁻¹ shear rate (arterial conditions). Our results showed higher fluorescence intensity, contraction force and stiffness of the microclots at 1500 s⁻¹, as compared to 500 s⁻¹. We also performed studies in the presence of HIP1 (anti-CD42b/GpIb α blocking antibody) at 1500 s⁻¹, and observed that HIP1 is more efficacious at the higher shears compared to 500 s⁻¹. This is consistent with the notion that VWF-GpIb α binding is more critical for platelet adhesion at arterial shears. In comparison, the anti-GpIIb/IIIa mAb Abciximab was an equally good blocker at both 500 and 1500 s⁻¹. These new results have been added to Fig 3 and Fig. 4 (main manuscript), and pg. 10 and 12 in the Results section in the main text.

3) *Even though there is novelty in the design of the collagen matrix, the measurements of clot contractile force and stiffness have previously been studied with post force sensors for platelets (Liang, Lab Chip, 2010) or whole blood (Judith, Lab Chip, 2015).*

We agree with this comment. While it is true that these previous studies allowed the measurement of clot contractile force and stiffness using micropost force sensors, the measurements were performed under static condition. In Liang, Lab Chip, 2010⁷, PRP was simply dropped on top of the micropost arrays and in Judith, Lab Chip, 2015⁸, whole blood sample filled a micro-fluidic chamber under capillary force (wicked). In both cases, the effects of shear flow on clot formation and mechanical property change were not studied. In contrast to these previous studies, our work clearly demonstrate an important role for fluid shear/hemodynamics on clot mechanics; therefore, we believe that our work can allow more

physiologically-relevant measurements that enable translation to the clinic. Furthermore, as mentioned by this reviewer, our microtissue contains a substantial amount of collagen that mimics exposed collagen upon vascular injury. This collagen substrate is missing in previous studies, as well as in clinical thromboelastography (TEG) devices.

4) It is not clear how consistent the collagen content is between individual microtissues within a clotMAT and between clotMATs, how dynamic platelet aggregates are on the microtissue, and how reproducible the assay is.

We thank the reviewer for this comment. The consistency in the collagen content is ensured by both the consistency in our microwell design and the good seeding practices. First, we formed the collagen microtissues using a well-established protocol similar to those used in our previous publications^{3, 4, 9}. The well-established protocol and good experimental practice help keep the microtissue quality consistent. Secondly, in the clotMAT device, each pair of micropillars is contained in a single microwell that receives collagen + HUVEC mixture during seeding process. Since the microwell volume is a constant and the collagen + HUVEC mixture composition (3mg/ml collagen and around 300 HUVECs per microwell) is also a constant, the collagen deposited in each microwell is a constant. As a result, after cell-mediated compaction and decellularization, the sizes of the dog-bone shaped collagen formed in each microwell (suspended between two pillars) is quite consistent. The measurement on collagen microtissue geometry showed that the collagen content is dog-bone shaped and is $132.7 \pm 17.2 \mu\text{m}$ wide (113 samples, 15 devices) and $80.7 \pm 5.9 \mu\text{m}$ thick (16 samples, 3 devices) before flow. Therefore, it represents a sufficiently large sample size (Supplementary Fig. 3). The small standard deviation suggests that the variation in collagen microtissue geometry is minimum and the level of consistency aligns with the good practice in the fields of microfabrication and tissue engineering. We have explicitly added these measurement in support of protocol reproducibility in Supplementary Figure 3.

In the clotMAT system, microclots form dynamically through gradually increased platelet adhesion over a 30 mins time period, as demonstrated by the continuous, temporal increase in fluorescence intensity (Fig. 3 and Supplementary movie 2). During this period, the mechanical properties (contractile force, stiffness) of the microclot continued to increase (Fig. 3), suggesting mechanobiological activation of the platelet.

Regarding reproducibility, we observed good consistency in the microclots formed within each clotMAT device (one donor). However, between different donors, the variation in microclot properties can be noticeable. Such variation is contributed by the inherent heterogeneity in donor population. Our statistical analysis based on a sufficiently large sample size (at least three donors per treatment condition with at least seven samples per donor) shows that the variation (standard deviation) is small as compared to the averaged value (mean) of the measurement. We have explicitly added the measurement of microclot dimensions in Supplementary Figure 5 and 6.

5) The diagnostic potential of clotMATs has not been adequately demonstrated, as only one type of vWf D type IIA plasma was tested, and this was tested at a suboptimal low shear rate (500s-1). The diagnostic potential may be enhanced if this device can also detect the effects of anti-platelet drugs (eg aspirin), or hyper reactive platelets (eg platelets from patients with cardiovascular disease) or thrombocytopenic blood.

We thank the reviewer for helpful suggestions. We have performed new experiments to further

demonstrate the diagnostic potential of the clotMAT device. As suggested by the reviewer, we studied the effects of aspirin (acetylsalicylic acid, ASA) on microclot formation and observed an inhibitory effect on microclot fluorescence intensity and contractile forces. These new data have been added to Fig. 4. We agree that the clinical relevance of the current device can be further expanded by performing tests with hyper reactive platelets from patients with cardiovascular diseases, or reagents used in common thromboelastography assays such as thromboplastin, heparinase and ellagic acid. We have included these points in pg. 18 of the Discussion section, and hope to pursue them as part of future clinical investigations in our laboratories.

Specific comments:

6) *Figure 1, it would be very helpful to illustrate all dimensions for all the features within a clotMAT: micropillars, microclots, top PDMS channel.*

We thank the reviewer for this valuable suggestion. We added the dimensions of the clotMAT system including micropillars, microclots and the top PDMS channel in Supplementary Figure 2.

It would be also useful and important to describe platelet aggregate formation in a more detailed manner, since “submillimeter-size or thrombi (100um)’ were formed as stated in Results page 7, such size thrombi would likely disturb local flow and thus influence ongoing platelet adhesion, especially if the height of top PDMS channel is limited. How dynamic are the platelet aggregates? How reproducible is platelet aggregate formation between pillars, between donors, between clotMATs?

We thank the reviewer for this comment. We have answered a similar question regarding microclot structure and size in Reviewer #1 question 1, but we are happy to answer it here again. We have performed new SEM imaging and confocal fluorescence microscopy to characterize the structural composition of our microtissues and included new results in the main text and supplementary figures. We observed that the microtissue consists of a dog bone-shaped collagen core covered by a thick shell of clot made of platelet and fibrin. Confocal reconstructed cross-sectional view of the microtissue shows that the clot shell thickness ($28.2 \pm 7.6 \mu\text{m}$) is approximately 78% the thickness of the collagen core ($36 \mu\text{m}$) (Results section in main text, Supplementary Fig. 5, 6). This clot layer thickness is consistent with previously reported thickness of the clot formed at the vessel injury site in a mouse model ¹. Since the height of the flow channel used in the current study is $500 \mu\text{m}$, the microclot will not typically significantly disturb the flow. However, we noted massive thrombus formation when exogenous thrombin was added and under these instances, indeed it is possible to occlude the flow (Supplementary Fig 21).

We have answered questions regarding dynamic platelet aggregation and reproducibility in previous General Comment’s question #4. Please refer to the answers in the earlier section.

Furthermore, taking advantage of HUVEC contraction force, the authors used an innovative approach to pre-stress the microtissue by seeding the HUVEC then removing them with trypsinization. It seems that 1800 nN (as described in the Method) is critical for clotMATs improved performance over the bare-collagen setup and presumably other existing micro-fluidic devices. The authors need to provide the variability and range of such pre-stressing levels and the explain their effect on the clotting process.

We thank the reviewer for this suggestion. The pre-stress contractile force of collagen

microtissues was $1.86 \pm 0.56 \mu\text{N}$, which was measured on 153 microtissues before flow experiments. We have added this new measurement in the “Microtissue contractile force measurement section” of the Method section. The contractile force generated by microclots after 30 mins of PRP flow at 500 s^{-1} (standard experimental condition) is around $6 \mu\text{N}$, which is much higher than the pre-stress force before flow; therefore, the small variation in the pre-stress contractile force does not significantly affect the clotting process or clot mechanical properties.

Fig 1D, the collagen content appeared even in individual microtissues. Can the collagen content be quantitated for individual microtissues to ensure a consistent level of collagen, as it will greatly influence platelet aggregation (extent and size), thus force and stiffness?

We thank the reviewer for this comment. A similar question has been answered in General Comment’s question #4 regarding the consistency of the collagen content. We are happy to answer it here again. The consistency in the collagen content is ensured by both the consistency in our microwell design and the good seeding practices. First, we formed the collagen microtissues using a well-established protocol similar to those used in our previous publications^{3, 4, 9}. The well-established protocol and good experimental practice helps to keep the microtissue quality consistent. Second, in the clotMAT device, each pair of micropillars is contained in a single microwell that receives collagen + HUVEC mixture during seeding process. Since the microwell volume is a constant and the collagen + HUVEC mixture composition (3mg/ml collagen and around 300 HUVECs per microwell) is also a constant, the collagen deposited in each microwell is a constant. As a result, after cell-mediated compaction and decellularization, the size of the dog-bone shaped collagen formed in each microwell (suspended between two pillars) is quite consistent. The measurement on collagen microtissue geometry showed that the collagen content is dog-bone shaped and is $132.7 \pm 17.2 \mu\text{m}$ wide and $80.7 \pm 5.9 \mu\text{m}$ thick before flow. The small standard deviation suggests that the variation in collagen microtissue geometry is minimum and the level of consistency aligns with the good practices in the fields of microfabrication and tissue engineering. We have explicitly added these measurement results in the Supplementary Figure 3.

Fig 1F-G, since anticoagulated PRP is used, why do the authors see fibrin mesh in the microclots?

The anticoagulant used here is citrate, and the collagen matrix can activate platelet under these conditions (Supplementary Fig. 19). This likely results in local thrombin release that promotes fibrin formation. While tissue factor is absent in this system, it is possible that elements of the contact pathway also contribute to fibrin formation. This is discussed on pg. 16-17 of the Revised Discussion.

7) Figure 2, the stretching membrane allows the detection of stiffness. Does it undermine contractile force measurement (pillar bending) (page 8)?

No, the stretching membrane does not undermine the contractile force measurement. In the clotMAT device, the middle layer (PDMS substrate containing micropillars) and bottom stretchable membrane layer were bonded through irreversible covalent bonds; therefore, the bottom layer acts like a thickened substrate and does not interfere with the bending of the micropillars when measuring spontaneous contractile force of the microclots (non-stretched condition). During stiffness measurement, an external stretch was initiated from the stretchable membrane and was transferred to the microclot through the micropillars. Micropillar bending reflects the amount of external tension applied to the

microclot by the membrane stretching; so micropillar and substrate membrane work together to enable stiffness measurement in this case.

8) Figure 3 shows a shear-dependent change in contractile force, stiffness and platelet aggregate size. These findings validate the sensitivity and accuracy of the detection systems using clotMATs, however it does not support (at least not fully) the shear contribution to the mechanical properties, since these changes are more likely intrinsic, or greatly influenced by the number of platelets adherent/delivered at the indicated time and shear rates.

We agree with the reviewer that the changes in microclot mechanical properties (contractile force, stiffness and size) presented in Fig. 3 are contributed not only by the shear flow but also by the number of adhered platelets. However, since the number of adhered platelets is regulated by the shear flow, as shown by the shear rate-dependent microclot fluorescence intensity in Fig. 3C, shear rate is the fundamental/initial cause for the subsequent changes in microclot properties. The effects of platelet binding and cell activation are also separable as illustrated in Fig. 4 B-E, where microclots were first formed before perfusion of either thrombin or ADP into the flow cell. Here, addition of both procoagulation treatments increase clot stiffness and contractility, without change in the number of platelets bound to the collagen substrate. To address the reviewer comment, we have revised the text on pg. 10-11 in the “Microclot mechanics under various shear flow conditions” section to clarify the contributions of shear flow and platelet number on the change of microclot properties.

9) Figure 4 is interesting that ADP (10 μ M) had a greater impact than thrombin (5 U/ml) on clot mechanical properties. Any explanation? Are platelets on microclots fully activated? With such high concentration thrombin, the platelets are likely to become procoagulant and lose their adhesive and contractile function. Alternatively, the anticoagulant remaining in the PRP may diminish the platelet-activating effects of thrombin? If this is case, the thrombin receptor activation peptide would be preferable.

We would like to first clarify that the ADP and thrombin flow experiments performed in Fig. 4 were in the absence of platelets or fibrinogen in the flow. Here, the microclots were pre-formed under regular PRP flow conditions before ADP/Thrombin were introduced in normal PBS buffer. Due to the substantial thickness and mass (25 μ m thick) of the formed clot shell in the microclot, the dominant mode of solute transport inside the clot is diffusion^{10, 11}. Therefore, smaller agonists (e.g. ADP, wt 427 Da) will diffuse faster and penetrate deeper into the core of the microclots, thus eliciting more embedded platelets to contract, than larger agonists (e.g. thrombin, wt 37 kDa). As a result, the overall contractile force of ADP treated microclots is higher than that treated with thrombin within the relatively short observation window of our flow experiments (10 mins). We have added this explanation in pg. 11-12 of the “Microclot mechanics under platelet agonist and antagonist treatments” section of the Results section.

To address this reviewer’s comment, we also performed studies where ADP and thrombin were mixed with citrated PRP just prior to the clotMAT flow experiment. Here, thrombin elicited a very strong clotting response that essentially blocked the flow channel within 10 mins of flow. Although, ADP also caused the formation of larger microclots as compared to non-treated PRP flow condition, these microclots are not large enough to block the channel (Supplementary Fig. 21).

We thank the reviewer for the suggestion on thrombin receptor activation peptide. We have performed new PRP flow experiments using a different anticoagulant PPACK and using Thrombin

Receptor Activator Peptide (TRAP-6) as agonist. Here, in the presence of PPACK and physiological calcium levels, we observed more platelet binding and higher contractile force generation in the early stages (first 10 mins) of the flow experiment, compared to studies where citrate was anticoagulant. Fluorescence intensity plateaued and then decreased and contractile force also decreased in the mid to late stage (10 – 30 mins) of the flow experiment when using PPACK, due to platelet detachment/emboli. This may be due to the reduced levels of fibrin in this system, as PPACK is a potent thrombin inhibitor. Formation of platelet deposits in the PPACK study in the presence of TRAP-6, however, rescued the decrease in fluorescence intensity due to stabilization of the platelet thrombi. This treatment, however, did not affect the contractile force (Supplementary Fig. 14). We have added these new results in pg. 9-10 of the Results section. Please note that, in addition to the clotMAT assay, various measurements of platelet activation with different anti-coagulants and stimuli (Supplementary Fig. 16) and measurements of fibrin formation (Supplementary Fig. 17) have also been performed to support the above assertions.

Fig 4B and 4F need a control. The impact of Ak2 in Fig F-J seemed mild, in contrast to the importance of VWF (shown in Fig 5). Is it due to low shear rates used (500s⁻¹), and platelet adhesion is not fully GPIb-VWF dependent at 500s⁻¹. High shear rates should be tested using AK2 treated platelets.

Yes, this is due to the effect of shear rate. Anti-GPIb α blocking mAbs completely block platelet deposition at 1500/s but are only partially effective at lower shear rates (500/s). This is due to the role of GpIIb/IIIa binding interactions that support platelet adhesion at all shears, with GPIb α -VWF binding being more critical at higher flows. Please note that all blocking data are being shown using anti-GPIb α clone HIP1 in the manuscript, not mAb AK2 as stated in the original submission. Both mAbs are functionally equivalent, but the wrong clone name was stated in the original submission. We have included these new experimental data in the revised Fig. 4 (main manuscript) and pg. 12 of the Results section.

The authors should re-evaluate the relative impacts of anti-GPIba, anti-integrin and anti-myosin treatments on platelet aggregation. Under the shear conditions employed, would anti-GPIba significantly reduce platelets recruitment? Blebbistatin certainly shouldn't. To my surprise, the impact of these treatments on this system is opposite to what would be expected. AK2 treatment had more platelet fluorescence than the other two; while blebbistatin had a profound effect on platelet aggregate size (4G,J), rather than selectively on contractile force and stiffness. What is the explanation?

We thank the reviewer for this suggestion. As mentioned in the above response, we have performed new experiments to test the anti-GPIb α antibody (HIP1 clone) under high shear rate (1500 s⁻¹) and the results using this reagent showed much improved inhibition of platelet adhesion and clot mechanical properties at 1500s⁻¹, compared to lower shear rates (500 s⁻¹), confirming the shear rate-dependent nature of GPIb α -VWF interactions. We have included these new data in pg. 12 in the Results section.

In terms of the effect of blebbistatin, previous study¹² Fig. 5 has shown that blebbistatin-treated platelets were able to adhere and form large aggregates on type I fibrillar collagen substrate; however, since the subsequent tight packing of platelets did not occur due to loss of contractility, the structure of the platelet thrombi was highly unstable. This lack of thrombus consolidation resulted in continual detachment of platelets from the thrombus surface within 5 mins of shear flow. We believe that a similar case occurred in our experiment – platelets first adhered to the collagen surface but because of low contractility-induced weak packing, thromboembolism occurred over the 30 mins flow period, leading to

reduced amount of adhered platelets and reduced mechanical properties of the clot. We have added this explanation on pg. 13 in the Results section.

10) Figure 5, it is unclear how the experiments were performed based on information provided in the figure legends and methods. It is thus hard to follow the results. Do the authors use normal platelets resuspended in vWD or normal plasma prior to perfusion through microMATs? If so, what do the “Washed platelet controls” mean in 5D, E, F?

Yes, we resuspend normal human washed platelets in either VWD plasma or normal healthy plasma in these studies. PRP prepared by one-step centrifugation is used as control. We have revised Fig. 5 and Methods pg. 21 to clarify the presentation. Specifically, we refer to washed platelets suspended in VWD plasma as “reconstituted VWD PRP (rVWD PRP)”, washed platelets suspended in normal healthy plasma as “reconstituted healthy PRP (rHealthy PRP)”, and normal PRP as “Healthy PRP”.

Does the reconstituted PRP in vWD plasma show defects in ristocetin-induced platelet aggregation?

Yes, we performed shear induced platelet activation (SIPAct) assays in the new Supplemental Fig. 22D using a cone-plate viscometer. This is similar to the ristocetin based assay, only more physiologically relevant. Indeed, here we observed lower platelet activation in studies using VWD plasma compared to healthy control.

Were the vWD type IIA plasma from a single patient, or pooled? If pooled, how many patients were included?

The VWD type 2A plasma was from a single donor based on supplier data. We have performed characterization by ourselves to validate the VWD plasma, as presented in the Supplementary data Fig. 22.

11) Since anti-coagulated PRP is used, how much fibrin is generated in the assay? This is critical, as fibrin is a major component of clot stiffness/contraction in vivo.

We have performed new SEM imaging and confocal fluorescence microscopy to characterize the fibrin content in our microtissues and included new results in the main text and supplementary figures. We observe that the microclot formed under citrated PRP flow consists of a dog bone-shaped collagen core covered by a thick shell of clot made of platelet and fibrin, as demonstrated by the co-localization of platelet signal and fibrin signal in the clot shell (Supplementary Fig. 5, 6). Confocal reconstructed cross-sectional view of the microtissue shows that the clot shell thickness ($28.2 \pm 7.6 \mu\text{m}$) is approximately 78% the thickness of the collagen core ($36 \mu\text{m}$) (Results section in main text, Supplementary Fig. 5, 6). This clot layer thickness is consistent with previously reported thickness of the clot formed at the vessel injury site in a mouse model¹. We have added these new results in pg. 7 in the Results section. Furthermore, we observed massive fibrin-rich thrombus formation when exogenous thrombin was added to citrated PRP flow and under these instances, it is possible to occlude the flow (Supplementary Fig 21). Similar effect was also achieved in recalcified citrate PRP flow where the formation of a large, fibrin-rich clot blocked the channel after 10 mins of flow (Supplementary Fig. 14). We have added these new results in pg. 9 and 12 in the Results section.

Please note that, in addition to the clotMAT assay, various measurements of fibrin formation with different anti-coagulants and stimuli have been performed under static condition using a plate reader (Supplementary Fig. 17).

12) At the highest shear rate (500s⁻¹) used in this study, the platelet aggregate size kept increasing over a 30 min perfusion period. Is this relevant to physiological haemostatic responses?

We would like to first clarify that the number of platelets increases with shear rate and time. However, the microclot size itself decreasing over 30 mins perfusion period as shown in Fig. 3B. This is due to clot retraction observed both in vivo¹ and in vitro^{13,14} under shear flow condition.

13) The increase in clot contractile force most likely comes from the increased number of platelets adhered to collagen with increasing shear as shown in Fig 2C. However, the tissue stiffness with increasing shear must take into account the stiffening behaviour of fibrin in response to shear (Litvinov Matrix Biol, 2016). Please comment.

We agree with the reviewer that the effect of shear force/straining on fibrin stiffness has been demonstrated in previous studies including the work by Litvinov et al¹⁵. In these studies, stiffening occurs when a fairly large strain was applied to induce large structural change of the fibrin. For example, the stiffening of fibrin fibers was observed at large strains >110% with an increase of the elastic modulus by a factor of 1.9 for crosslinked and 3.0 for uncrosslinked fibers, respectively¹⁶. In the current study, we showed that under platelet antagonists treatment (abciximab and HIP1 in Fig. 4 G-I, platelet adhesion is minimum), shear flow alone did not cause obvious change in microclot size or mechanical properties, suggesting that shear force generated by the PRP flow is probably too low to induce the large strain needed for matrix stiffening. This may be due to the relatively large volume and pre-stress of the microtissue as compared to the low level shear force generated by the flow. As a comparison, straining/deformation of the fibrin presented in previous studies was induced using compression and shear rheometer, which can generate quite large force/strain. Another evidence is that control experiment performed using PBS flow (no platelet, no antagonist or agonist) did not cause any change in the size or contractile force of the collagen microtissue (supplementary Fig.9), suggesting the fluid shear force is minor as compared to the inherent mechanical parameters of the microtissue.

Other minor comments:

1. It would be useful to show a low magnification fluorescence image revealing platelet aggregates on multiple pillar heads to demonstrate reproducibility within a device, since the platelet deposition seemed heterogeneous on collagen microtissue (Fig 2B, Fig 3B,E, Fig 5B).

We thank the review for this suggestion. We already have a low magnification image in Fig. 1C-E to show the microclots formed within one device. As it can be seen, the microclot geometries are quite consistent. As we mentioned previously, the major heterogeneity is due to donor to donor variation, rather than intra-donor variability. Such heterogeneity has, however, been taken into account in our statistical analysis.

2. *It is very impressive for the authors' team to make micro-pillar system for measuring clot retraction. Based on Fig.S4, it seems the aspect ratio of the micro-pillars are >7:1 (height:width). With such high aspect ratios, the pillars are very difficult to preserve and remain functional if damaged. It will be useful to provide more insight into how these technical challenges have been overcome.*

We are happy to provide explanation on our techniques to any interested scientist. To overcome this technical challenge, we treated the master mold with saline to render the PDMS surface non-adhesive, which will allow the micropillars to be easily demolded. Glass micro-pipettes were used to gently straighten the micropillars if they fell after demolding.

3. *The measurement of clot stiffness by membrane stretching is important. The information on how mechanical force is loaded and the force loading data are missing.*

We thank the reviewer for this comment. We have added representative force loading data in Supplementary Fig. S10.

4. *Do the authors have any insights whether this device is suitable for mouse blood studies?*

We have not tried the clotMAT system with mouse blood sample. However, based on the system response, we expect it will allow the clotting of mouse blood sample and the measurement of clot properties. Mouse platelets are however smaller than human platelets, and a direct comparison may or may not be possible.

5. *Page 10, line 2. It mentions healthy donor's microclot measured in the paper is comparable to thrombi collected from stroke patients (26 kPa). This is potentially interesting, but how is this related?*

We would like to first clarify that it is ADP-treated healthy donor's microclot that has similar stiffness (26 kPa) as comparable to thrombi collected from stroke patients. Clinical studies^{17, 18} have shown that ADP-induced platelet aggregation is associated with the outcome of the cardiovascular diseases. For example, it has been shown that platelet hyperreactivity to ADP is significantly associated with incident myocardial infarction/stroke¹⁷. Therefore, we performed ADP-treatment on the microclot and compared its stiffness to published stiffness of thrombi collected from stroke patient, which is known to be stiff.

6. *The dimension of clotMats need to be more obviously illustrated (pillar leg height: 100um; pillar heads, collagen microtissue: 100um, top PDMS channel dimension).*

We thank the reviewer for this comment. More detailed dimensions have been added to the clotMAT system schematics in the Supplementary Fig. 2.

References

1. Stalker, T.J. et al. Hierarchical organization in the hemostatic response and its relationship to the platelet signaling network. *Blood*, blood-2012-2009-457739 (2013).
2. Zhu, S., Lu, Y., Sinno, T. & Diamond, S. Analysis of morphology of platelet aggregates formed on collagen under laminar blood flow. *J Biol Chem* **291**, 23027-23035 (2016).

3. Zhao, R., Boudou, T., Wang, W.G., Chen, C.S. & Reich, D.H. Decoupling cell and matrix mechanics in engineered microtissues using magnetically actuated microcantilevers. *Advanced Materials* **25**, 1699-1705 (2013).
4. Zhao, R., Chen, C.S. & Reich, D.H. Force-driven evolution of mesoscale structure in engineered 3D microtissues and the modulation of tissue stiffening. *Biomaterials* **35**, 5056-5064 (2014).
5. Liu, A.S. et al. Matrix viscoplasticity and its shielding by active mechanics in microtissue models: Experiments and mathematical modeling. *Scientific reports* **6**, 33919 (2016).
6. Muthard, R.W. & Diamond, S.L. Blood clots are rapidly assembled hemodynamic sensors: flow arrest triggers intraluminal thrombus contraction. *Arteriosclerosis, thrombosis, and vascular biology* **32**, 2938-2945 (2012).
7. Liang, X., Han, S., Reems, J., Gao, D. & Sniadecki, N. Platelet retraction force measurements using flexible post force sensors. *Lab Chip* **10**, 991 (2010).
8. Judith, R.M. et al. Micro-elastometry on whole blood clots using actuated surface-attached posts (ASAPs). *Lab on a Chip* **15**, 1385-1393 (2015).
9. Asmani, M. et al. Fibrotic microtissue array to predict anti-fibrosis drug efficacy. *Nat Commun* **9**, 2066 (2018).
10. Diamond, S.L. & Anand, S. Inner clot diffusion and permeation during fibrinolysis. *Biophysical journal* **65**, 2622-2643 (1993).
11. Blinc, A. et al. Lysing patterns of retracted blood clots with diffusion or bulk flow transport of plasma with urokinase into clots—a magnetic resonance imaging study in vitro. *Thrombosis and haemostasis* **68**, 667-671 (1992).
12. Ono, A. et al. Identification of a fibrin-independent platelet contractile mechanism regulating primary hemostasis and thrombus growth. *Blood* **112**, 90-99 (2008).
13. Myers, D. et al. Single-platelet nanomechanics measured by high-throughput cytometry. *Nat Mater* **16**, 230 (2017).
14. Sakurai, Y. et al. A microengineered vascularized bleeding model that integrates the principal components of hemostasis. *Nature communications* **9**, 509 (2018).
15. Litvinov, R.I. & Weisel, J.W. Fibrin mechanical properties and their structural origins. *Matrix Biology* **60**, 110-123 (2017).
16. Liu, W., Carlisle, C., Sparks, E. & Guthold, M. The mechanical properties of single fibrin fibers. *Journal of Thrombosis and Haemostasis* **8**, 1030-1036 (2010).
17. Puurunen, M.K. et al. ADP platelet hyperreactivity predicts cardiovascular disease in the FHS (Framingham Heart Study). *Journal of the American Heart Association* **7**, e008522 (2018).
18. Cha, J.K., Jeon, H.W. & Kang, M.J. ADP - induced platelet aggregation in acute ischemic stroke patients on aspirin therapy. *European journal of neurology* **15**, 1304-1308 (2008).

REVIEWERS' COMMENTS:

Reviewer #1 (Remarks to the Author):

Authors addressed the issues raised in the review with new and appropriate experiments that validate the system. While many devices and studies have explored clot mechanics on whole blood clots or PRP clotted under static conditions, those studies have clots that are completely different from those formed under flow. This important study is the first to measure clot mechanics when the clots actually are formed under flow as happens in the body.

Reviewer #2 (Remarks to the Author):

The original manuscript described a new and innovatively designed device called clotMATs. The device represents an important technical advance in modelling in vivo clot formation, and a useful research and potential diagnostic tool to detect various bleeding disorders. The main concerns with the original manuscript related to 1) the use of citrated PRP, rather than whole blood, to detect clot mechanical properties; 2) the studies were performed over a narrow shear range (3-500s⁻¹); 3) there was insufficient clarity on clotMATs dimensions; 4) concerns over assay consistency and dynamics; and 5) issues related to the diagnostic potential of this device.

The authors have done an excellent job revising their manuscript. They have performed a substantial number of additional experiments, modified several of the main figures, added many more supplementary figures, revising the original manuscript and providing additional explanation and discussion. Most of the specific concerns were also adequately addressed, except for one piece of data related to fibrin formation.

The authors detect a substantial amount of fibrin following perfusion of citrate PRP through clotMATs (Fig 1G, and new Supp Fig 6, and in Author's response to Specific Concerns point 11 and 13). This raises two queries. 1) Fibrin formation should not be occurring during the perfusion of citrate PRP. If citrate anticoagulation is inadequate, the assay would be highly variable as the collected blood/PRP would generate small amounts of thrombin over time. 2) The forming thrombin/fibrin would be expected to influence clot stiffness, contraction force and/or thrombus size. However, the new data using PPACK-PRP (in Supp Fig 15) showed no difference in contractile force between citrate RPR and PPACK-PRP, although the clot stability (size) was reduced in PPACK-PRP at later perfusion times. Does this suggest that the contractile force is mainly or solely mediated by platelets, not fibrin in this assay system under the employed experimental conditions? However, the authors stated on page 17, line-370-372 "contraction is observed even when PPACK is anti-coagulant suggesting that at least some of the contractile force is transmitted to the micropillars via adhesive interactions between platelets and collagen substrate, presumably via receptors GpVI and VLA-2'. The authors should state more clearly whether fibrin contributes to contractile force in this assay system.

Minor points:

1, page 16: 367-368: PPACK is a thrombin inhibitor, not 'thrombin receptor antagonist'.

Shaun Jackson

We appreciate the time and effort each of the reviewers have dedicated to providing insightful feedback on our manuscript. The following section provides our point-by-point response, and outlines the changes made to the manuscript. Reviewer comments appears in italicized text, followed by our response in red font.

Reviewer #1 (Remarks to the Author)

Authors addressed the issues raised in the review with new and appropriate experiments that validate the system. While many devices and studies have explored clot mechanics on whole blood clots or PRP clotted under static conditions, those studies have clots that are completely different from those formed under flow. This important study is the first to measure clot mechanics when the clots actually are formed under flow as happens in the body.

We appreciate the positive comments from this reviewer.

Reviewer #2 (Remarks to the Author):

The original manuscript described a new and innovatively designed device called clotMATs. The device represents an important technical advance in modelling in vivo clot formation, and a useful research and potential diagnostic tool to detect various bleeding disorders. The main concerns with the original manuscript related to 1) the use of citrated PRP, rather than whole blood, to detect clot mechanical properties; 2) the studies were performed over a narrow shear range (3-500s⁻¹); 3) there was insufficient clarity on clotMATs dimensions; 4) concerns over assay consistency and dynamics; and 5) issues related to the diagnostic potential of this device.

The authors have done an excellent job revising their manuscript. They have performed a substantial number of additional experiments, modified several of the main figures, added many more supplementary figures, revising the original manuscript and providing additional explanation and discussion. Most of the specific concerns were also adequately addressed, except for one piece of data related to fibrin formation.

We thank the reviewer for the above comments.

1. The authors detect a substantial amount of fibrin following perfusion of citrate PRP through clotMATs (Fig 1G, and new Supp Fig 6, and in Author's response to Specific Concerns point 11 and 13). This raises two queries. 1) Fibrin formation should not be occurring during the perfusion of citrate PRP. If citrate anticoagulation is inadequate, the assay would be highly variable as the collected blood/PRP would generate small amounts of thrombin over time. 2) The forming thrombin/fibrin would be expected to influence clot stiffness, contraction force and/or thrombus size. However, the new data using PPACK-PRP (in Supp Fig 15) showed no difference in contractile force between citrate RPR and PPACK-PRP, although the clot stability (size) was reduced in PPACK-PRP at later perfusion times. Does this suggest that the contractile force is mainly or solely mediated by platelets, not fibrin in this assay system under the employed experimental conditions? However, the authors stated on page 17, line-370-372 "contraction is observed even when PPACK is anti-coagulant suggesting that at least some of the contractile force is transmitted to the micropillars via adhesive interactions between platelets and collagen substrate,

presumably via receptors GpVI and VLA-2'. The authors should state more clearly whether fibrin contributes to contractile force in this assay system.

Yes, indeed, we observed substantial fibrin formation in our assay even when using 3.4% citrate as anti-coagulant. This is likely due to thrombin released locally at the collagen-platelet interface following platelet activation as stated in the manuscript, rather than a failure of the anti-coagulation strategy. Consistent with this notion, the microclot-fibrin mesh only forms on the collagen substrate in the proximity of platelets, and not at other places in the flow channel (Fig. 1 C-E, Supplementary Movie 1). Thus, this is a spatially-temporally controlled process. The fibrin formed is sufficient to support a platelet clot and result in retraction forces locally, but not excessive to the extent that it blocks the flow within the 30 min experiment window. Thus, the clotMAT assay could be a viable method in future clinical applications.

With regard to the relative importance of fibrin in this system with respect to contractile force, our studies with PPACK suggest that clot contraction is possible even when fibrin formation is reduced. The contractile force upon use of this anti-coagulant is, however, ~2 μ N less compared to citrated PRP at the 30 min time point (Supplementary Fig. 15). This observation suggests that force transmission to the clotMAT pillars is likely primarily due to platelet-collagen interactions, presumably via receptors GpVI and VLA-2. Fibrin may play a relatively small role in direct force transmission in this assay. However, the platelet clots formed when using PPACK were unstable and they embolized, suggesting that fibrin plays an important indirect role by stabilizing the platelet thrombi even if its role as a force transducer is relatively small. The text in Discussion has been edited to clarify this point.

We thank the reviewer for providing incisive critique that has helped us to improve the study design and presentation of results.

Minor points:

1, page 16: 367-368: PPACK is a thrombin inhibitor, not 'thrombin receptor antagonist'.

We have made the text correction as suggested.